# SHIP164 is a chorein motif lipid transfer protein that controls endosome–Golgi membrane traffic

Michael G. Hanna[1,2,3,4]* , Patreece H. Suen[2]* , Yumei Wu[1,2,3,4], Karin M. Reinisch[2,6] , and Pietro De Camilli[1,2,3,4,5,6]

**Cellular membranes differ in protein and lipid composition as well as in the protein–lipid ratio. Thus, progression of membranous organelles along traffic routes requires mechanisms to control bilayer lipid chemistry and their abundance relative to proteins. The recent structural and functional characterization of VPS13-family proteins has suggested a mechanism through which lipids can be transferred in bulk from one membrane to another at membrane contact sites, and thus independently of vesicular traffic. Here, we show that SHIP164 (UHRF1BP1L) shares structural and lipid transfer properties with these proteins and is localized on a subpopulation of vesicle clusters in the early endocytic pathway whose membrane cargo includes the cation-independent mannose-6-phosphate receptor (MPR). Loss of SHIP164 disrupts retrograde traffic of these organelles to the Golgi complex. Our findings raise the possibility that bulk transfer of lipids to endocytic membranes may play a role in their traffic.**

## Introduction

The homeostasis of intracellular membranes and their adaptation to changes in the functional state of the cell requires the coordination of protein and lipid transport. While much has been learned about protein traffic, less is known about the dynamics and transport of bilayer lipids. A significant fraction of such lipids moves between organelles as part of the membranes of vesicular carriers. However, lipids also move via transport proteins that harbor them in hydrophobic cavities as they travel through the aqueous environment of the cytosol. This mode of transport has been known for decades but has been increasingly appreciated over the last several years with the discovery of many new lipid transport proteins. Moreover, it has also become clear that many such proteins function at membrane contact sites, thus facilitating specificity and speed of lipid transport. Typically, these proteins contain modules or motifs that tether them to the two apposed membranes, while lipid transfer modules move back and forth between them to extract and deliver lipids by a shuttling mechanism (Alva and Lupas, 2016; Prinz et al., 2020; Saheki and Camilli, 2017; Reinisch and Prinz, 2021; Wong et al., 2019).

Recently, the characterization of VPS13 and its distant paralog ATG2 has suggested a new mode of transport involving the flow of lipids along a protein bridge that connects the two membranes in eukaryotic cells (Kumar et al., 2018; Valverde et al., 2019; Li et al., 2020; Leonzino et al., 2021; Noda, 2021).

A defining feature of these proteins is the presence of a conserved N-terminal region, ~125 residues long, referred to as the chorein-N motif (Kumar et al., 2018; Osawa et al., 2019; Ueno et al., 2001; Rampoldi et al., 2001). In VPS13 and ATG2, this motif caps one end of an elongated rod (extended chorein domain). The rod comprises an extended β-sheet that is highly curved to resemble a taco shell harboring a groove along its length (Li et al., 2020; Valverde et al., 2019). A hydrophobic cavity in the chorein motif is continuous with the groove, whose floor is lined by hydrophobic amino acids, and so suited to accommodate many lipids at once and to allow their flow from one end of the rod to the other. In ATG2, the rod represents the bulk of the protein, while in VPS13 there are additional C-terminal domains that function in localization. Lipids are thought to flow unidirectionally along the rod, producing a net flow of lipids to the acceptor membrane and allowing for its expansion independent of contribution of new membrane lipids by vesicle fusion (Leonzino et al., 2021). Accordingly, both the VPS13 paralogs and ATG2 have been implicated in membrane growth (Park et al., 2013; Da Costa et al., 2020; Mari et al., 2010; Chang et al., 2021). Many questions remain, however, about the precise mechanisms of action of these proteins. In some cases, the earliest stages of membrane growth mediated by VPS13 and ATG2 are characterized by the presence of clusters of small vesicles, raising questions about a potential interplay between the lipid

[1]Department of Neuroscience, Yale University School of Medicine, New Haven, CT;   [2]Department of Cell Biology, Yale University School of Medicine, New Haven, CT;   [3]Howard Hughes Medical Institute, Yale University School of Medicine, New Haven, CT;   [4]Program in Cellular Neuroscience, Neurodegeneration and Repair, Yale University School of Medicine, New Haven, CT;   [5]Kavli Institue for Neuroscience, Yale University School of Medicine, New Haven, CT;   [6]Aligning Science Across Parkinson's Collaborative Research Network, Chevy Chase, MD.

*M.G. Hanna and P.H. Suen contributed equally to this paper.   Correspondence to Pietro De Camilli: pietro.decamilli@yale.edu;   Karin M. Reinisch: karin.reinisch@yale.edu.

transport properties of these proteins and vesicle fusion (Park et al., 2013; Da Costa et al., 2020; Mari et al., 2010). As studies of VPS13 have informed regarding ATG2 function and vice versa, characterization of other family members should yield further insights both as to how these proteins function and the cellular processes in which they participate.

A predicted chorein motif is present at the N-terminus of SHIP164 (also called UHRF1BP1L) and in its paralog UHRF1BP1 (also called C6orf107), and fold prediction algorithms indicate high confidence that downstream portions form an extended chorein domain β-sheet (Jumper et al., 2021; Yang et al., 2020; Fig. 1 A). SHIP164 was first identified as an interactor of the Habc domain of Syntaxin 6 (Stx6) and shown to be a component of a multimolecular assembly including, subunits of the Golgi-associated retrograde protein (GARP) complex (Otto et al., 2010). GARP is a tethering complex, which along with the structurally similar endosome-associated recycling protein (EARP) complex cooperates with Stx6 in the fusion of endocytic vesicles with acceptor membranes in endosomes (EARP) and the Golgi complex (GARP; Pérez-Victoria and Bonifacino, 2009; Schindler et al., 2015; Abascal-Palacios et al., 2013). Moreover, over-expressed SHIP164–Stx6 complex was shown to colocalize with the cation-independent mannose-6-phosphate receptor (MPR) and perturb the traffic of this receptor (Otto et al., 2010). More recently, SHIP164 was identified as a top hit in a cellular-based screen for Rab5 effectors, further supporting a role of SHIP164 in the early endocytic pathway (Gillingham et al., 2019). The possibility that SHIP164 may represent a lipid transport protein suggests that its characterization may reveal novel aspects of the biology of the endosomal system. Additional interest in this protein comes from its identification as a Parkinson's disease candidate gene from a large-scale whole-exome sequencing study (Jansen et al., 2017). To date, nothing is known about the cell biology of UHRF1BP1.

The goal of this study was to test the hypothesis that SHIP164 may be a lipid transport protein and to gain new information about its localization and physiological function. Here, we provide direct evidence for a structural similarity of SHIP164 to ATG2 and the N-terminal half of VPS13 and for the property of SHIP164 to harbor and transport lipids. We show that SHIP64 localizes to clusters of small vesicles in the endocytic pathway and demonstrate its importance in retrograde traffic to the Golgi complex by both over-expression and loss-of-function studies.

## Results
### Molecular properties of SHIP164 support a lipid transport function
As a first step in assessing whether SHIP164 might function as a lipid transporter in the VPS13 family, we assayed whether it can solubilize and transport lipids between membranes, and, if so, which ones. We expressed full-length, N-terminally FLAG-tagged human SHIP164 (3xFLAG-SHIP164) in Expi293 cells and purified it using anti-FLAG affinity resin, amid extensive washing to remove non-specifically bound lipids, then analyzed the sample for co-purified molecules by shotgun mass spectrometry lipidomics (Fig. 1 B). A large assortment of lipids,

primarily phospholipids, were co-purified with SHIP164. There was no significant enrichment of a particular phospholipid relative to the content of the whole cell, however.

A construct lacking a long-predicted unstructured segment (residues 901–1099) of SHIP164 was used in subsequent in vitro studies to improve protein solubility. Further supporting that SHIP164 binds phospholipids, 3XFLAG-SHIP164$_{\Delta901-1099}$ binds nitrobenzoxadiazole (NBD)-labeled phosphatidylethanolamine (PE), co-migrating with this lipid on a native gel (Fig. 1 C). By comparing the fluorescence that co-migrated with SHIP164 versus a well-characterized lipid transport module from the protein E-Syt2, we estimated that each SHIP164 molecule can accommodate multiple lipids (~8), as might be expected if SHIP164 has an extended lipid-binding groove like VPS13 or ATG2 (Kumar et al., 2018; Valverde et al., 2019).

We took advantage of the ability of SHIP164 to bind NBD-PE in designing an in vitro FRET-based assay to monitor lipid transfer between membranes (Kumar et al., 2018). In this assay, SHIP164 is tethered (Fig. 1 D) between donor liposomes containing both rhodamine (Rh)- and NBD-labeled PE and acceptor liposomes, initially lacking these fluorescent lipids. Förster resonance energy transfer (FRET) between Rh and NBD reduces NBD fluorescence initially. Transfer to acceptor liposomes and consequent dilution of the fluorescently tagged lipids would reduce FRET and lead to increased NBD fluorescence. Consistent with a role in lipid transfer, addition of SHIP164, but not an "empty" tether lacking a lipid transport module, led to increased fluorescence (Fig. 1 E). Addition of SHIP164 to donor liposomes only, in the absence of acceptor liposomes, led to a fast, smaller fluorescence increase due to the ability of SHIP164 to extract lipids from the liposomes (Fig. 1 E, red arrow), causing some dilution (similar to observations for ATG2 and VPS13; Kumar et al., 2018; Valverde et al., 2019). The possibility that the increased fluorescence is due to SHIP164-mediated fusion between donor and acceptor liposomes was ruled out using a dithionite quenching assay (Fig. 1 F). These experiments support that SHIP164 can solubilize lipids and transfer them between membranes.

We next examined SHIP164 by electron microscopy. 2D class averages of negatively stained 3XFLAG-SHIP164$_{\Delta901-1099}$ revealed a 200-Å-long rod (Figs. 1 G and S1 A). Comparison of SHIP164 with an MBP-SHIP164 fusion indicates that this rod represents a tail-to-tail dimer as there are two densities corresponding to maltose-binding protein (MBP), one at each end. The MBP-tag locates the SHIP164 N-terminus at the rod end. Whether this dimerization is physiologically relevant is unclear; for example, ATG2 similarly dimerizes in vitro, at high concentrations, and in the absence of binding partners, but is thought to be monomeric in vivo (Valverde et al., 2019). Further, characterization of 3XFLAG-SHIP164$_{\Delta901-1099}$ using single particle cryo-electron microscopy techniques yielded a reconstruction at an estimated resolution of ~8.3 Å (Fig. S1, B and C). As suggested by fold prediction algorithms (Jumper et al., 2021; Yang et al., 2020; Fig. 1 A), a cavity runs along the entire length of the rod, as in VPS13 and ATG2 (Fig. 1, H and I). The algorithms also predict that this cavity is lined entirely with hydrophobic residues (Fig. 1 A [2]). Thus, the cavity can accommodate the

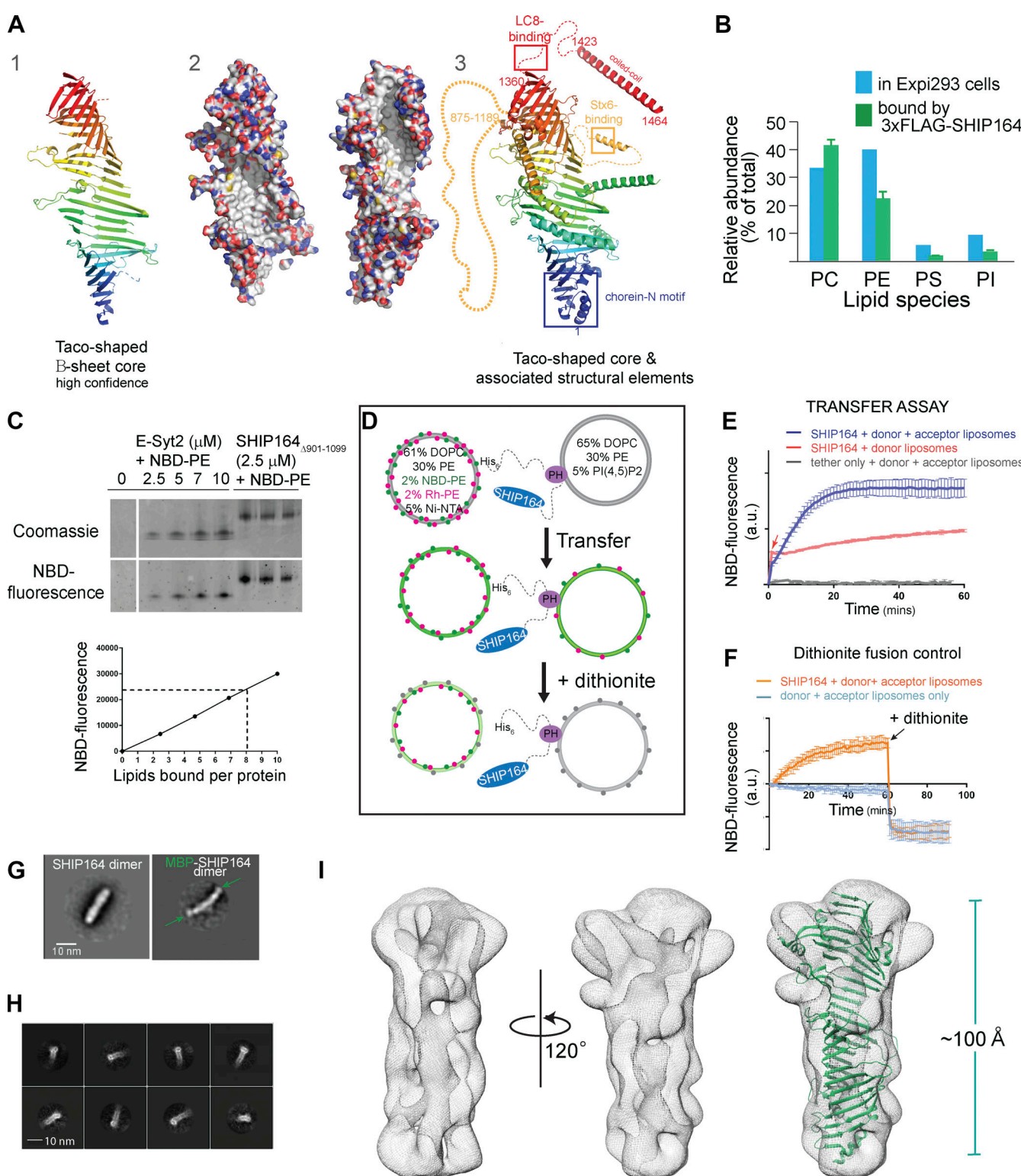

Figure 1. **In vitro characterization of SHIP164. (A)** The fold prediction algorithm AlphaFold indicates that SHIP164 harbors an extended β-sheet folded to resemble a taco shell, as in VPS13 (Li et al., 2020). From left to right: (1) Ribbons representation for the taco shell "core," predicted with high confidence. (2) The extended β-sheet is shown in two orientations as surface representation, with carbons shown white, oxygens red, nitrogens blue, and sulfurs yellow. The concave surface of the taco shell is predicted to be entirely hydrophobic, providing a binding site for lipid fatty acid moieties. (3) Additional secondary structure that may associate with the core is shown; segments discussed in the text are indicated. **(B)** SHIP164 co-purifies with phospholipids according to their a-bundance in cells as assessed by shotgun lipidomics. **(C)** SHIP164 co-migrates with NBD-PE in a native gel. Comparison of NBD fluorescence that co-migrates with SHIP164 versus with the well-characterized lipid transport module from E-Syt2 suggests that each SHIP164 molecule accommodates multiple (~8) phospholipids. **(D)** Schematic drawing explaining the FRET-based lipid transfer and the dithionite-based control for the absence of liposome fusion. **(E)** Assay consistent with transfer of fluorescent lipids from donor to acceptor liposomes by SHIP164 but not by a tether construct lacking a lipid transfer module. An

increase in fluorescence observed with donor liposomes only (red arrow) is due to lipid extraction by SHIP164. A larger increase is observed when lipids are subsequently transferred to the acceptor liposomes. **(F)** Dithionite addition excludes that SHIP164 facilitates membrane fusion, as the fluorescence decrease is the same in the SHIP164 sample and the no-protein control. In the case of protein-mediated fusion, NBD fluorescence would be increased with respect to the no-protein control. **(G)** 2D class averages from negatively stained SHIP164 and MBP-SHIP164, showing tail-to-tail dimerization. MBPs (arrows) are at the ends of the elongated dimer. **(H)** 2D class averages for the cryo-EM reconstruction show a channel running along the length of SHIP164. **(I)** Cryo-EM map of SHIP164 at a resolution of ∼8.3 Å, showing a long central cavity with or without the AlphaFold model (see A 1) superimposed onto it. The docking is approximate because the resolution of the reconstruction is low and because a high confidence model for SHIP164, beyond the β-sheet taco shell core, is not available. Experiments in E and F were performed in triplicate; SDs are indicated. mins, minutes; PC, phosphatidylcholine; PI, phosphatidylinisitol; PS, phosphatidylserine. Source data are available for this figure: SourceData F1.

---

multiple lipids bound by SHIP164, and we propose that it could be a conduit for lipids to transit between membranes.

Collectively, these findings are consistent with the hypothesis that SHIP164 is a lipid transport protein. We next investigated its site of action within cells.

### Localization of exogenous SHIP164 points to a role on endocytic organelles

Antibodies directed against SHIP164 did not yield a consistently reliable signal when tested by immunofluorescence. Thus, as a first step toward the identification of the site of action of SHIP164, we examined the localization of exogenously tagged human SHIP164. As studies of VPS13 had shown that tags appended to the N-terminus (i.e., the N-terminus of the chorein motif) interfere with its physiological function (Kumar et al., 2018; Park et al., 2016), we engineered tags at the C-terminus of SHIP164 (SHIP164-Halo) or an internal site within the predicted disordered region (after amino acid residue 915; SHIP164^mScarlet; Kumar et al., 2018; Park et al., 2016; Fig. 2 A). When either of these tagged SHIP164 constructs was expressed alone, punctate structures of varying size were observed throughout the cytoplasm, with the larger and brighter spots localized in the central region of the cell, in proximity of the Golgi complex area (Fig. 2 A). Conversely, even a small N-terminal tag appended to SHIP164^mScarlet (3xFLAG-SHIP164^mScarlet) abolished the presence of cytoplasmic foci (Fig. S2 A; Otto et al., 2010).

The scattered distribution of SHIP164^mScarlet foci throughout the cytosol, combined with previous results (Otto et al., 2010) suggesting a role of SHIP164 in the endocytic pathway, prompted us to examine the colocalization of SHIP164 relative to endosome markers. Many, but not all, SHIP164 puncta localized in proximity of spots or vacuoles positive for GFP-2xHrs^FYVE, a PI3P binding probe (Stenmark et al., 2002; Fig. S2 B). Moreover, SHIP164^mScarlet spots were observed in close proximity to other co-expressed markers of endosome subcompartments (Zoncu et al., 2009; Fig. 2 B). SHIP164, however, did not localize on the organelles positive for these probes, but on foci juxtaposed to them (Fig. 2 B). In some cases, foci were localized at sites where the endosomes appeared to be close to the ER (Fig. 2 C) but not restricted to the space between these two organelles. As shown by live-cell imaging, SHIP164 foci not only were variable in size as well as in fluorescence intensity but were also highly dynamic structures. They changed shape, often underwent fission, or coalesced into larger spots, although they tended to remain tethered to endosomes and micropinosomes (Fig. 2 D and Video 1).

A more precise colocalization of exogenous SHIP164 was observed with Rab5 (BFP-Rab5b), confirming previous results (Otto et al., 2010; Gillingham et al., 2019; Fig. 2 B). In this case, a pool of SHIP164 colocalized with Rab5 along the entire profile of vacuoles and, conversely, Rab5 robustly colocalized with the bright SHIP164 foci closely as opposed to large vacuoles (Fig. 2 B). However, expression of a dominant negative Rab5 mutant (GFP-Rab5a^S34N) did not abolish bright SHIP164 (SHIP164-Halo) foci, suggesting that Rab5 is not necessary for their formation (Fig. S2 C).

Notably, exogenous SHIP164 puncta colocalized with endogenous MPR, and its expression (leading to higher-than-normal levels of SHIP164) resulted in an alteration of the localization of the MPR, which accumulated in large foci where the two proteins colocalized (Fig. 2 E). These accumulations occurred throughout the cell but were especially large in proximity of the Golgi complex, where the normal TGN-like appearance of MPR was replaced by these large accumulations located at some distance from GM130, a Golgi complex marker (Nakamura et al., 1995; Fig. 2 E). A similar redistribution of MPR and also of the Golgi/endosome protein sortilin (Lefrancois et al., 2003) was observed in RPE1 (retinal pigment epithelial) cells transfected with untagged SHIP164, ruling out non-specific effects of the engineered fluorescent tag of exogenous SHIP164 (Fig. S2, D–G). Such changes were not observed with TGN46, another protein enriched in the TGN that cycles to and from the plasma membrane and endosomes (Pérez-Victoria and Bonifacino, 2009; Luzio et al., 1990; Wakana et al., 2012; Fig. S2, H and I). This difference is in line with the known traffic of MPR and sortilin versus TGN46 in different vesicular carriers (Petersen et al., 1997; Wakana et al., 2012; see also below).

### Foci of exogenous SHIP164 reflect accumulations of small vesicles

To gain further insight into the precise nature of foci of exogenous tagged SHIP164, we performed correlative light–electron microscopy (CLEM) of COS-7 cells co-expressing the early endosome marker GFP-WDFY2 and SHIP164^mScarlet (Fig. 3 A). Both conventional transmission electron microscopy (TEM) and focused ion beam scanning electron microscopy (FIB-SEM) were performed. We employed Ras^G12V to induce the formation of macropinosomes (Porat-Shliom et al., 2008) to facilitate the alignment of fluorescence and EM images. This analysis revealed that spots of SHIP164 fluorescence reflected accumulations of hundreds of tightly packed ∼50–60-nm vesicles juxtaposed to endosomal membranes positive for GFP-WDFY2 (Fig. 3 A), thus providing an explanation for the dynamic nature

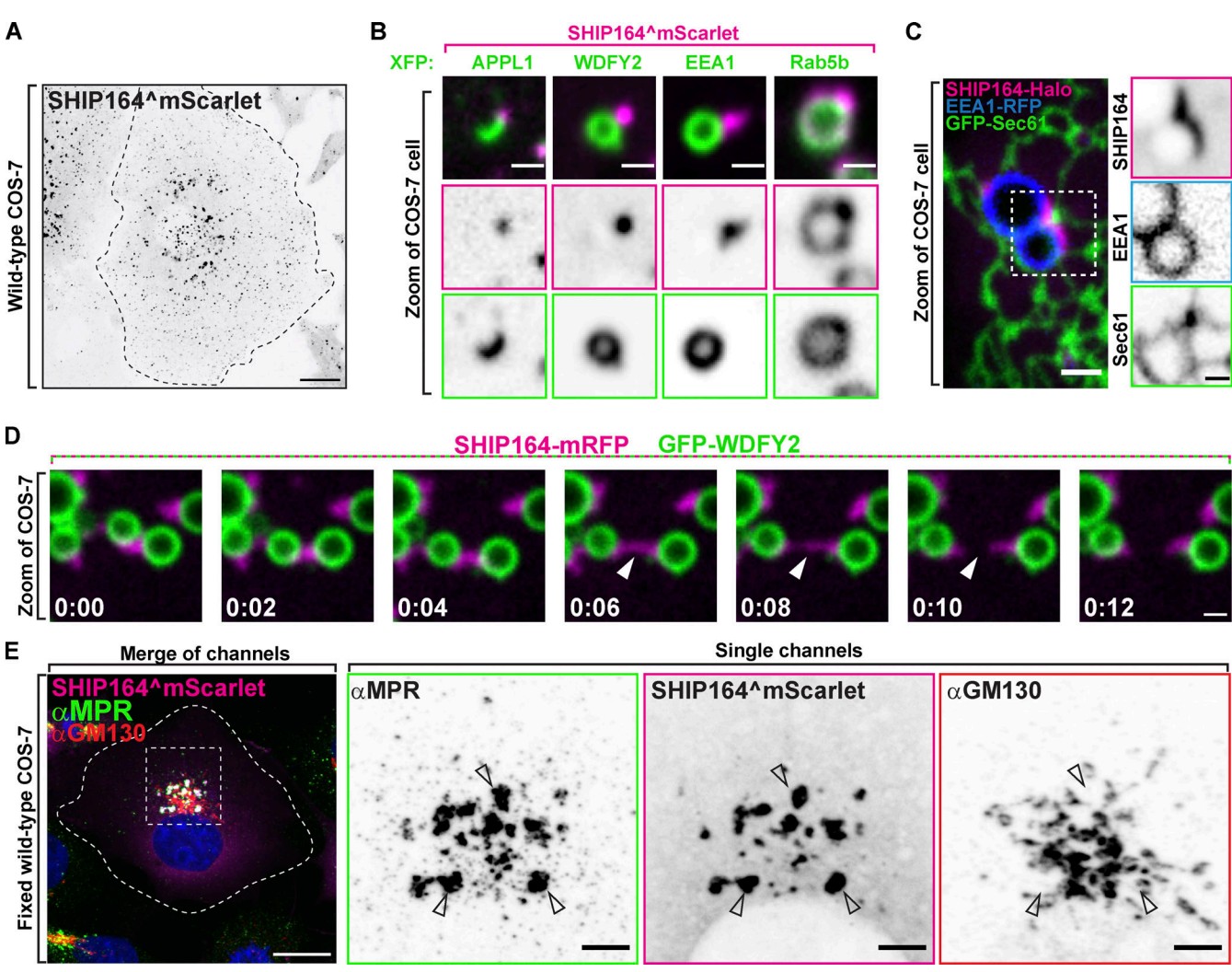

Figure 2. **Localization of exogenous SHIP164 points to a role on endocytic organelles. (A)** Live fluorescence (inverted grays) image of a COS-7 cell expressing exogenous SHIP164^mScarlet. Scale bar, 20 μm. **(B)** High-magnification live fluorescence images of COS-7 cells expressing exogenous SHIP164^mScarlet (magenta) and the organelle marker indicated at the top (green). The individual channels are shown as inverted grays. Scale bar, 2 μm. **(C)** Live image of the cytoplasm of a COS-7 cell expressing exogenous SHIP164-Halo (magenta), the endosome marker EEA1-RFP (blue), and the ER marker GFP-Sec61 (green). Scale bar, 2 μm. The individual channels are shown as inverted grays. Scale bar, 1 μm. **(D)** Time-series of live fluorescence images of exogenous SHIP164-mRFP (magenta) and EEA1-GFP (green). Arrowheads point to a SHIP164 accumulation undergoing fission. Time, seconds. Scale bar, 1 μm. **(E)** Fluorescence images of a COS-7 cell expressing exogenous SHIP164^mScarlet (magenta) and immunolabeled with antibodies against MPR (green) and GM130 (red). Scale bar, 20 μm. The zoom of individual channels are shown as inverted grays. Arrowheads indicate colocalization of exogenous SHIP164 and endogenous MPR. Scale bar, 5 μm.

of the fluorescence spots. An ~70-nm-thick matrix appeared to anchor these clusters to the WDFY2-positive vacuoles, and the space occupied by this matrix was nearly devoid of vesicles (Fig. 3, A–C). ER tubules were present in proximity of these clusters and often penetrated them, but there did not seem to be sites of preferential accumulation of vesicles (Fig. 3 B). The mechanism responsible for the clustering of SHIP164 positive vesicles remains unclear. Also unclear is how such clusters are anchored to the large vacuoles.

Collectively, the findings described above, based on exogenous tagged SHIP164, strongly supported a role of SHIP164 in the endocytic pathway. However, large accumulation of small vesicles, such as those described here, were never observed to our knowledge in WT cells, making it critical to assess the localization of SHIP164 when expressed at the endogenous level.

**Endogenous SHIP164 localizes to small clusters of vesicles near the cell edge**

We tagged SHIP164 at the endogenous locus in HeLa cells. Specifically, we engineered a mNeonGreen (mNG) tag in the disordered region after amino acid residue 1092 via the ORANGE genomic editing method (Willems et al., 2020; Fig. 4 A). Two clones were isolated where expression of endogenous SHIP164^mNG (eSHIP164^mNG) was confirmed by the presence of a band running at the appropriate molecular weight and recognized by Western blotting with both anti-SHIP164 and anti-mNG antibodies (Fig. 4 B). Correct insertion of the tag was

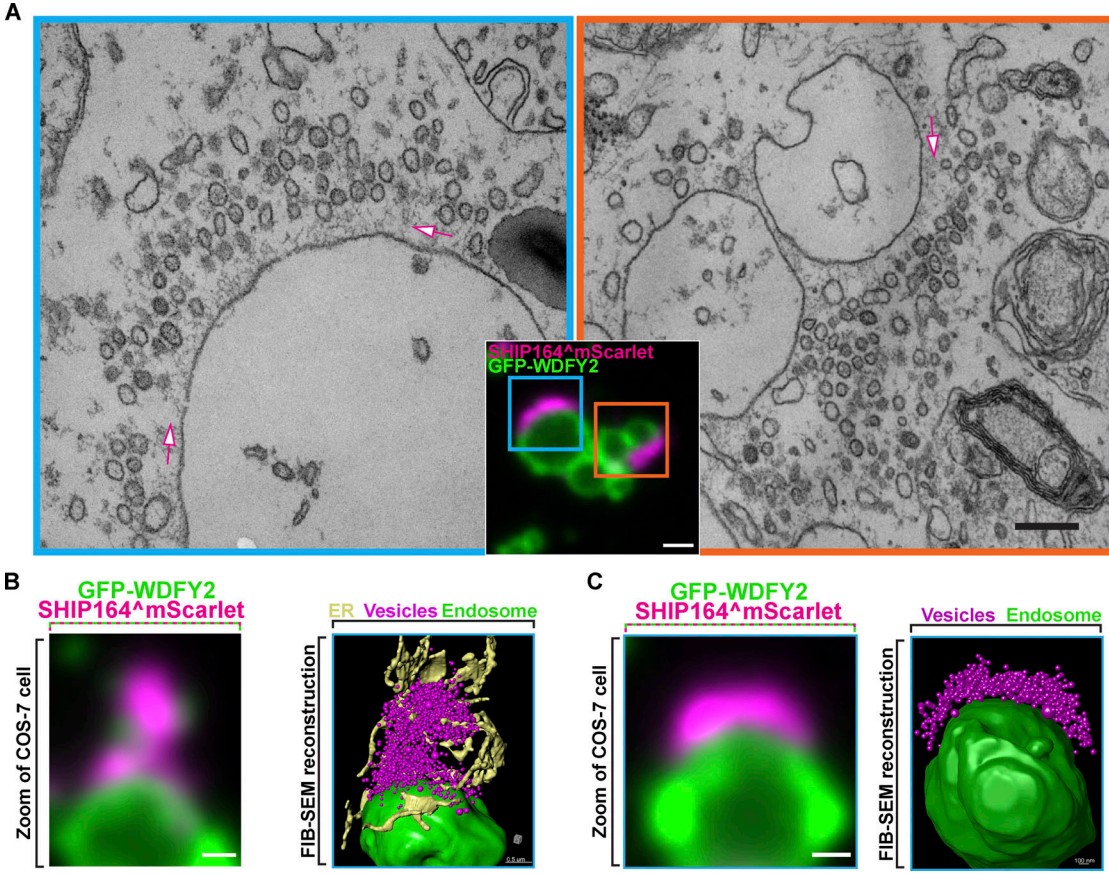

**Figure 3. CLEM reveals that foci of exogenous SHIP164 reflect large accumulations of small vesicles. (A)** SHIP164^mScarlet-positive structures juxtaposed to large GFP-WDFY2–positive endosome compartments in a COS-7 cell also expressing untagged Ras^G12V to induce macropinosomes. The two EM images (scale bar, 200 nm) correspond to the regions framed by two rectangles in the merge fluorescence image (scale bar, 1 μm). SHIP164 fluorescence reflects clusters of small vesicles separated from the endosomes by a band occupied by a dense matrix (arrows). **(B and C)** High-power view of endosomes of COS-7 cells expressing exogenous SHIP164^mScarlet (magenta), GFP-WDFY2 (green), and untagged Ras^G12V. Merge fluorescence images are shown at left and FIB-SEM–based reconstructions are shown at right. Scale bar, 1 μm.

validated by DNA sequencing of the targeted gene region (Fig. 4 C).

Like exogenous SHIP164, eSHIP164^mNG had a punctate appearance. However, eSHIP164^mNG puncta were very much smaller and, in contrast to foci of over-expressed SHIP164 (Fig. 2 A), were primarily localized at the cell edge (Fig. 4 D). On the other hand, when SHIP164^mScarlet was expressed in these endogenously tagged cells, exogenous mScarlet and endogenous mNG fluorescence precisely overlapped in large foci, including the ones localized in deep regions of the cell (Fig. S2 J). This demonstrates that endogenous and exogenous SHIP164 completely intermix—as expected from correct gene targeting— confirming that large foci are the result of SHIP164 over-expression.

To determine whether puncta of eSHIP164^mNG represent MPR positive vesicles, we generated a double knock-in HeLa cell line where both SHIP164 and MPR were tagged at their endogenous loci (eSHIP164^mNG and eMPR-mScarlet). Most, but not all, eSHIP164^mNG positive puncta colocalized and moved together with puncta of eMPR-mScarlet fluorescence (Fig. 4 E and Video 2). Conversely, many eMPR-mScarlet puncta at the edge of the cell colocalized and moved together with puncta of eSHIP164^mNG (Fig. 4 E), while MPR vesicles located deeper into the cell were negative for SHIP164 (Waguri et al., 2003; Lin et al., 2003). Fluorescent puncta positive for the two proteins were distinct from vesicles positive for newly internalized labeled EGF (EGF-647) revealing that endogenous SHIP164 localizes to a specific endocytic subpopulation (Fig. S2 K). Interestingly, an intrinsic membrane protein of unknown function, that is listed as a SHIP164 interactor in BioGRID (https://thebiogrid.org), FAM174a, strikingly colocalized with exogenous SHIP164 accumulations, although FAM174a was also present in the Golgi complex and other compartments of the secretory/endocytic pathways (Fig. S3, A and B).

To gain further insight into the nature of endogenous SHIP164 puncta, we again employed CLEM in eSHIP164^mNG HeLa cells, also expressing a marker of mitochondria (mito-BFP), to aid in fluorescence and EM image alignment (Fig. 4 F). Given the small size of the fluorescent structures, FIB-SEM rather than TEM was used in spite of the lower resolution afforded by FIB-SEM as this technique allows a better alignment between fluorescence and EM images in 3D. This analysis revealed that, similar to the large fluorescence puncta of exogenously expressed SHIP164, spots of eSHIP164^mNG fluorescence

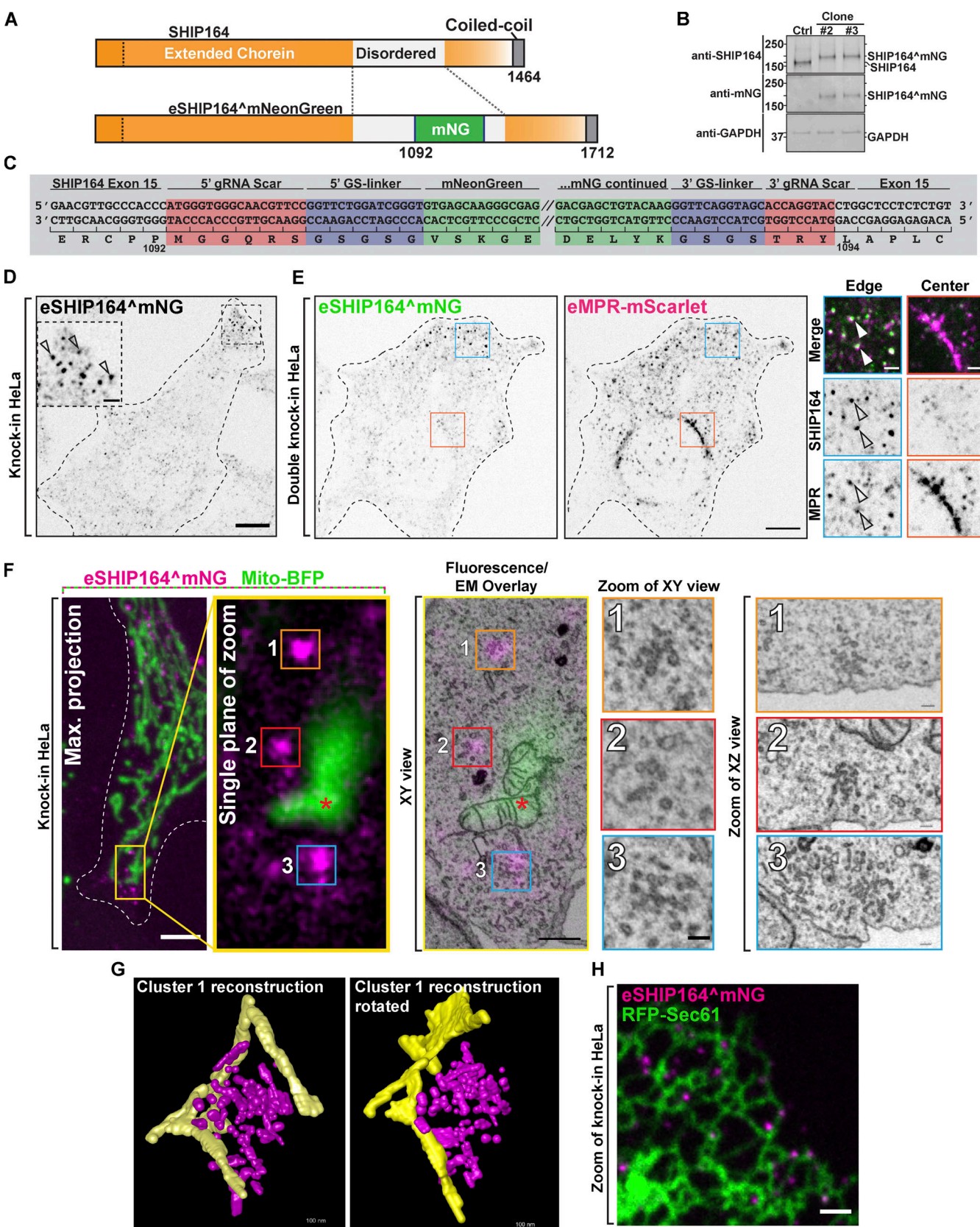

**Figure 4. Localization of Endogenous SHIP164 on small clusters of vesicular structures. (A)** Cartoon of WT (top) and edited SHIP164^mNG (bottom). **(B)** Western blot (in kD) of control and edited cell clones for SHIP164, mNG, and GAPDH as a loading control. The different motility of SHIP164 in control and edited clones indicates the incorporation of mNG in the edited SHIP164. **(C)** Cartoon of endogenously edited locus of SHIP164^mNG in HeLa cell. Red, DNA scar

as a result of homology-independent targeted insertion; blue, small Gly-Ser linkers; green, mNG sequence (shortened for brevity by forward slashes). **(D)** Fluorescence image of endogenous SHIP164^mNG (inverted grays) in an edited HeLa knock-in cell. Arrowheads indicate concentration of endogenous SHIP164 at the cell edge. Scale bar, 10 µm (main field) and 2 µm (inset). **(E)** Single channel (inverted grays) fluorescence images of endogenous SHIP164^mNG and endogenous MPR-mScarlet in a HeLa double-knock-in cell. Scale bar, 10 µm. High-magnification scale bar, 2 µm. **(F)** Left to right: Fixed fluorescence image of endogenously tagged SHIP164^mNG (magenta) HeLa cell also expressing mito-BFP (green) for alignment in CLEM analysis. Scale bar, 5 µm. Single plane of the area is outlined in yellow. Fluorescence overlay with FIB-SEM image: numbered boxes mark region of interest of eSHIP164^mNG fluorescence to be aligned with EM image. Scale bar of magnification area, 1 µm. Magnification of cluster of vesicles aligned with eSHIP164^mNG fluorescence marked by numbered boxes. Scale bar of EM area, 100 nm. **(G)** Reconstruction of cluster 1 from panel F above. SHIP164-associated endosome structures, magenta; ER tubules, yellow. **(H)** Fluorescence image of endogenously tagged SHIP164^mNG in a HeLa knock-in cell also expressing the ER marker RFP-Sec61 (green) demonstrating some SHIP164 foci are in proximity of ER tubules. Scale bar, 2 µm. Source data are available for this figure: SourceData F4.

reflected tightly packed clusters of small vesicles and short tubules (Fig. 4, F and G). However, these clusters were much smaller (tens instead of hundreds of vesicular structures). As revealed by 3D reconstruction of FIB-SEM volumes (Fig. 4 G) and by fluorescence of cells also expressing the ER marker RFP-Sec61 (Fig. 4 H), ER tubules were localized in close proximity of these clusters, but vesicles were preferentially associated with themselves rather than with the ER.

Collectively, these findings indicate that SHIP164, when expressed at endogenous levels, is selectively localized on small MPR-positive vesicles near the cell periphery. They further imply that SHIP164 dissociates from such vesicles as they travel to deeper cellular regions and that the presence of exogenous SHIP164 expands the size of the vesicle clusters in central cell regions near the Golgi complex. To better understand SHIP164's dynamics in the endocytic pathway, we investigated its interactions.

### Interaction of SHIP164 with proteins implicated in the targeting of endocytic cargoes to the Golgi complex

We confirmed the previously reported colocalization of exogenous SHIP164 with over-expressed Stx6 (Otto et al., 2010; Fig. 5 A). However, the location of the Stx6 binding site within the SHIP164 protein was not known. A short hydrophobic motif ($L_{37}xxYY_{41}$) responsible for binding to the Habc domain of Stx6 was previously identified in the N-terminal region of VPS51 (Abascal-Palacios et al., 2013). A similar motif ($L_{759}xxYY_{763}$) is present in a predicted disordered loop of SHIP164 within the extended chorein domain (Fig. 1 A [3]; and Fig. 5 E), and mutations of both tyrosine resides (Y772A, Y773A) abolished the recruitment of GFP-SHIP164 (GFP-SHIP164$^{Y772A, Y773A}$; i.e., N-terminally tagged SHIP164 that when expressed by itself is cytosolic) to mRFP-Stx6 (Fig. S3 C). However, the same mutations, when introduced into C-terminally tagged SHIP164 (SHIP164$^{Y772A, Y773A}$-mRFP), did not affect the localization of the protein into bright foci in cells not transfected with Stx6, further indicating that Stx6 does not account for these localizations (Fig. S3 D).

To identify additional SHIP164 interacting partners that may impact its localization and/or function, we performed pull-down experiments from detergent-solubilized mouse brain lysates onto purified SHIP164 (3xFLAG-SHIP164) bound to anti-FLAG resin. Shotgun mass spectrometry of the affinity-purified material showed significant specific enrichment (relative to control) of the light chains 1 and 2 (DYNLL1/2; also known as LC8) of cytoplasmic dynein, the minus-end directed microtubule motor

(Carter et al., 2016; Fig. 5 B). DYNLL1/2 act as dynein complex adaptors that mediate a wide variety of protein–protein interactions through a short linear binding motif (Rapali et al., 2011a). Accordingly, a motif that fits this consensus is present in SHIP164 near its C-terminus (Fig. 5 E), and LC8 is another protein listed among putative SHIP164 interactors in BioGRID. Moreover, when an MBP-tagged fragment of SHIP164 containing this region (MBP-SHIP164$_{1312-1464}$) and untagged LC8 were co-expressed in bacteria from a polycistronic plasmid, LC8 co-purified with MBP-SHIP164$_{1312-1464}$ on amylose-resin and co-fractionated with MBP-SHIP164$_{1312-1464}$ when the eluate of the resin was fractionated on a size-exclusion column, indicating a direct interaction (Fig. 5 C).

Interestingly, another potential SHIP164 interactor listed in BioGRID, Rab45 (RASEF), is linked to dynein function. Rab45 and its better-characterized paralog CRACR2a are dynein/dynactin activators that connect endocytic cargoes to minus-end microtubule transport (Wang et al., 2019; Shintani et al., 2007). In agreement with the possibility that SHIP164 may be an effector of Rab45, over-expression of Rab45 dramatically concentrated both transfected SHIP164^mScarlet and endogenously tagged SHIP164 (eSHIP164^mNG) to perinuclear spots (i.e., the position occupied by the centrosome), depleting them from the periphery (Figs. 5 D and S3 E). At this perinuclear location, SHIP164 and Rab45 precisely colocalized (Figs. 5 D and S3 E). Over-expression of Rab45 also clustered endogenous MPR tagged at the C-terminus with an mNG tag (eMPR-mNG), but only in cells over-expressing SHIP64, i.e., in cells where the majority of MPR-positive vesicles (not only the peripheral vesicles) are positive for SHIP164 (Fig. S3 F). A dominant-inactive Rab45 (S555N) mutant had a cytosolic localization and did not induce SHIP164 accumulation in this area suggesting that the nucleotide bound state of Rab45 regulates recruitment (Fig. S3 G).

Taken together, our interaction studies point to an association of SHIP164 with vesicles that traffic from the cell periphery to the centrosomal/Golgi complex region, although endogenous SHIP164 appears to be shed by vesicles as they move to central regions of the cells unless Rab45 is over-expressed. One would therefore expect a role of SHIP164 in some aspects of retrograde membrane traffic. To validate this hypothesis, we performed loss-of-function studies.

### Defects in retrograde membrane traffic to the TGN in SHIP164 knockout cells

To assess the impact of the absence of SHIP164 on cell physiology, we chose RPE1 cells as a model system due to their reliable

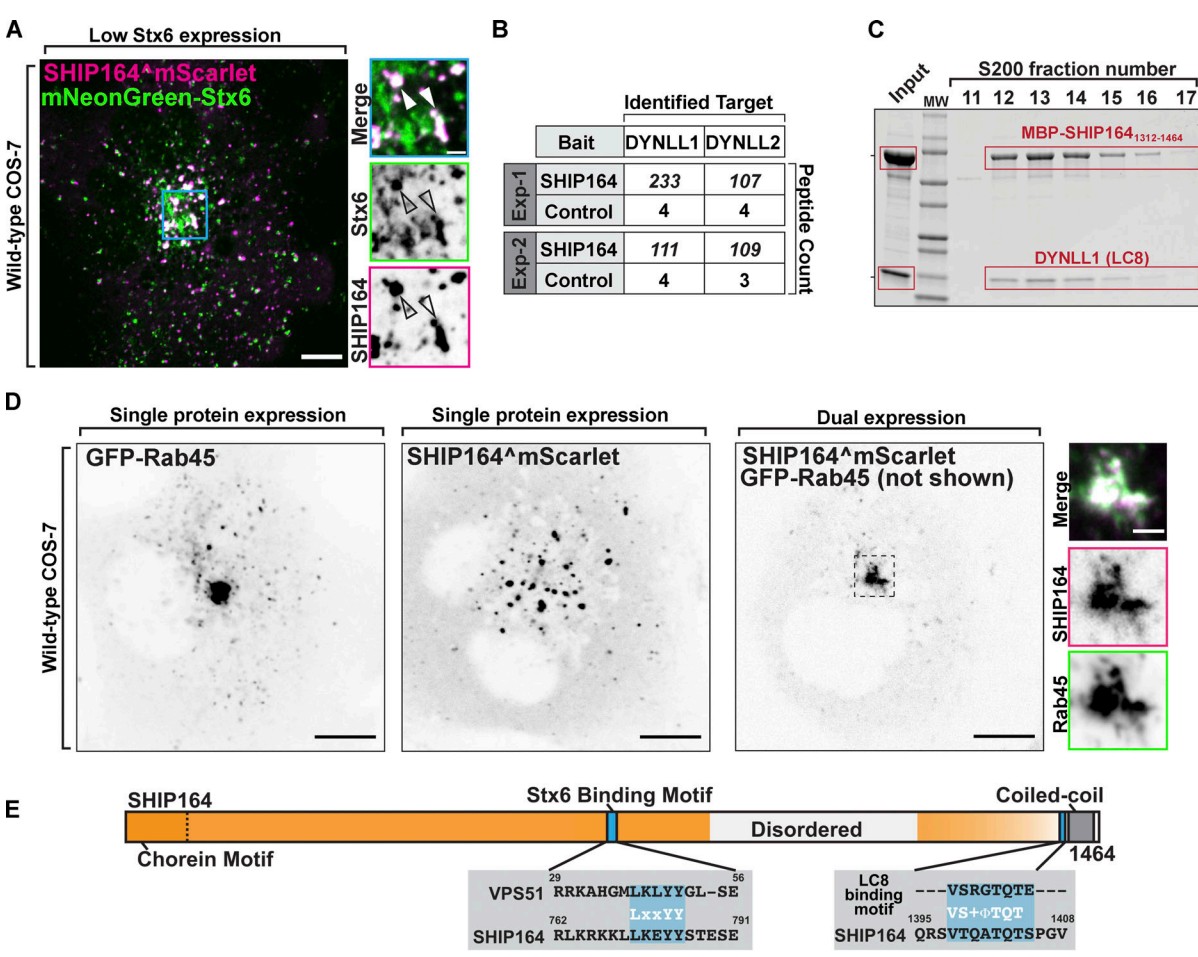

**Figure 5. Interaction of SHIP164 with proteins implicated in retrograde traffic to the Golgi complex. (A)** Live fluorescence image of a COS-7 cell demonstrating partial colocalization of exogenous SHIP164^mScarlet (magenta) and mNG-Stx6 (green). Scale bar, 10 μm. High magnification of the indicated regions is shown at right, and the individual channels are shown as inverted grays. Arrowheads indicate overlapping fluorescence. Scale bar, 2 μm. **(B)** Mass spectrometry–based identification of material affinity-purified onto immobilized SHIP164 or onto a control protein from mouse brain extract. The selective enrichment of DYNLL1/2 peptides on the SHIP164 bait is shown. **(C)** A mixture of MBP-SHIP164$_{1312-1464}$ and DYNLL1/2 was subjected to size exclusion chromatography. Coomassie Blue staining of SDS-PAGE of the eluted fractions reveals co-migration of the two proteins. **(D)** Live fluorescence images (inverted grays) of COS-7 cells expressing either GFP-Rab45 (left) or SHIP164^mScarlet (right), or both proteins together (only SHIP164 is shown) as indicated. Scale bar, 10 μm. High-magnification scale bar, 2 μm. **(E)** Cartoon of WT SHIP164 highlighting (blue) newly identified Stx6 and DYNLL1/2 interaction motifs (see also Fig. 1 A). Source data are available for this figure: SourceData F5.

genetic editing and flat morphology optimal for organelle imaging. Using CRISPR/Cas9 methodology, two independent SHIP164 knockout (KO) clones were generated. In both clones, identical indels were observed on both alleles (but different between the two clones), and absence of the SHIP164 protein was confirmed by Western blotting (Figs. S4 A and 6 A). No obvious changes were observed in cell shape, in mitochondria, ER, or Golgi morphology as assessed by the cis-Golgi marker (GM130; see below). Two differences, however, were noted.

One was a difference in the size of endosomes labeled by EEA1 (visualized by immunostaining), and GFP-2xHrs$^{FYVE}$ (visualized by live-cell imaging) as the largest (>1 μm$^2$) vesicles positive for these markers in WT cells were mostly absent in KO cells (Fig. 6, B and C; **, P < 0.01).

The other difference was a change in the steady-state localization of MPR and other proteins that recycle between the cell periphery (plasma membrane and endosomes) and the TGN.

Specifically, upon immunostaining of SHIP164 KO cells for the MPR, an increase in the pool of MPR localized in vesicles scattered throughout the cytoplasm relative to the pool localized in the Golgi complex area was observed (Lin et al., 2003), (Fig. 6 D; **, P < 0.01). The localizations by immunofluorescence of TGN46 and sortilin showed a similar difference relative to controls (Figs. 6 E and S4 B; **, P < 0.01), although the vesicles positive for TGN46 were different from those positive for MPR and sortilin (Fig. S4 C). As a control, the ratio of clathrin heavy chain immunoreactivity, which is sparse throughout the cytoplasm relative to immunoreactivity in the Golgi complex, was also measured, and this ratio was not statistically different between SHIP164 KO cells and parental control cells (Fig. 6 F).

To validate the dependence of these phenotypes on the lack of SHIP164, we attempted to rescue them by expressing exogenous SHIP164 in the KO cells. To this aim, we used the polycistronic construct described above encoding both untagged SHIP164, to

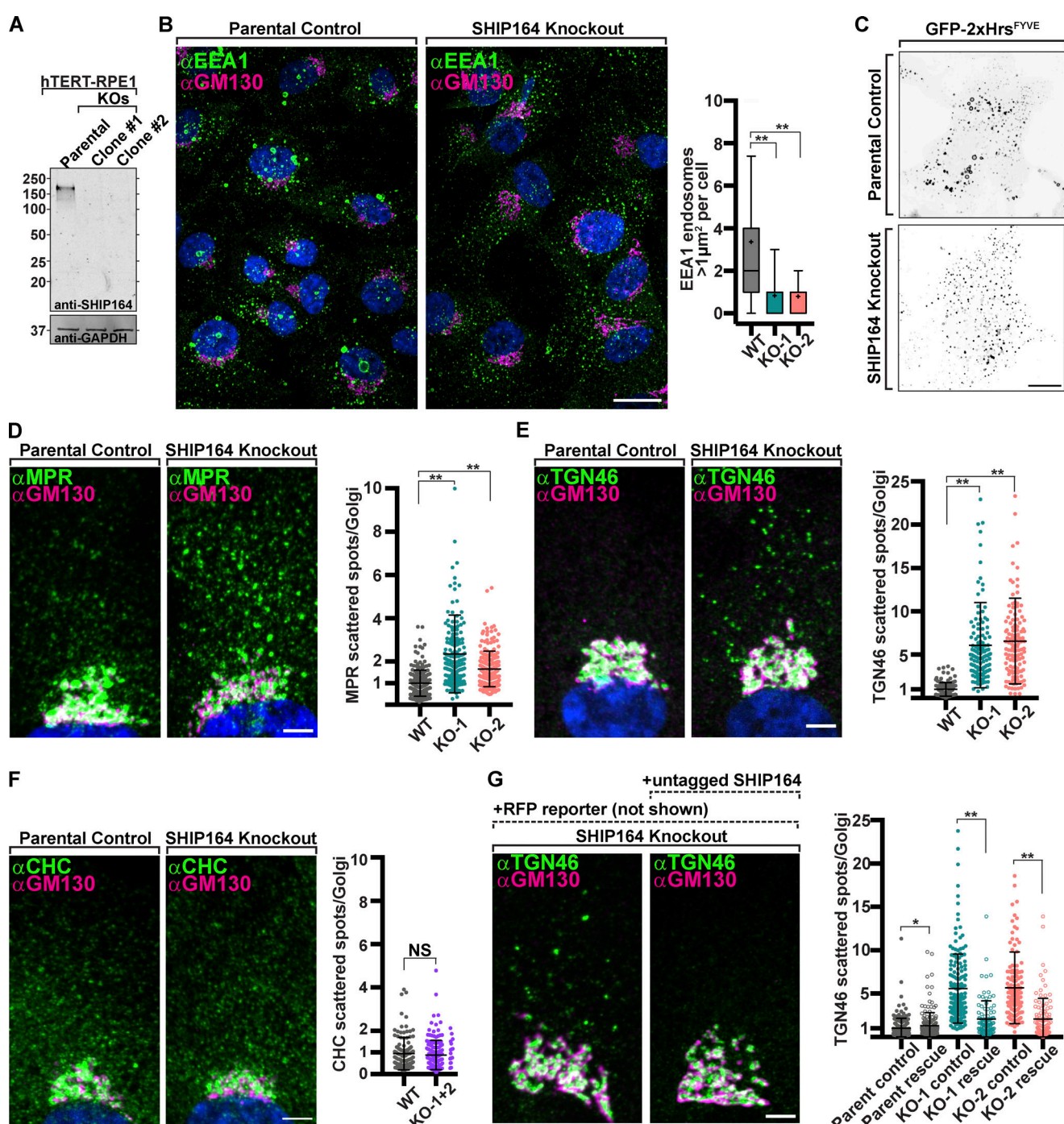

Figure 6. **Defects in retrograde membrane traffic to the TGN in SHIP164 knockout cells. (A)** Western blot (in kD) of control and edited RPE1 cell clones for SHIP164 and GAPDH as a loading control. **(B)** Left: Fluorescence images of a parental control (left) and SHIP164 KO cells (right) immunolabeled with antibodies against EEA1 (green) and GM130 (magenta). Scale bar, 20 μm. Right: Quantification of large EEA1 endosomes (>1 μm²) per cell in control and SHIP164 KO cell clones. **(C)** Live fluorescence (inverted grays) image of parental control (top) and SHIP164 KO (bottom) cells expressing the endosomal marker GFP-2xHrs[FYVE]. Scale bar, 20 μm. **(D–F)** Left: Fluorescence images of parental control (left) and SHIP164 KO cells (right) immunolabeled with antibodies against indicated protein (green) and GM130 (magenta). Scale bar, 5 μm. Right: Quantification of scattered cytoplasmic spots–to–Golgi complex ratio of indicated protein signal per cell in control and SHIP164 KO cell clones. **(G)** Left: Fluorescence images of SHIP164 KO cells expressing RFP alone (left) or both RFP and SHIP164 (right) immunolabeled with antibodies against TGN46 (green) and GM130 (magenta). Scale bar, 5 μm. Right: Quantification of scattered cytoplasmic spots–to–Golgi complex ratio of TGN46 signal per cell in SHIP164 KO cell clones expressing RFP alone (solid circles) or both RFP and SHIP164 (open circles). Data of B and D–F reflect three biological replicates and of G, two biological replicates. For B, D–F, and G: middle line, mean; bars, SD. **, P < 0.01; *, P < 0.5. Source data are available for this figure: SourceData F6.

avoid any artifact due to the tag, and soluble RFP to identify transfected cells. In both KO clones, transfection of this construct, but not of a construct encoding only the RFP reporter, resulted in the rescue of the vesicles-to-Golgi ratio of TGN46 immunoreactivity (Fig. 6 G; *, P < 0.05; **, P < 0.01). However, in the case of MPR and sortilin, expression of the SHIP164 construct did not restore their WT localization but resulted in their robust accumulation into large foci, as shown above upon SHIP164 over-expression in WT cells (Fig. S4 D). This result is most likely explained by the disrupting effect of SHIP164 over-expression, which, as shown above, does not impact TGN46 localization (Fig. S2, H and I). Likewise, SHIP164 over-expression did not seem to rescue the presence of the large EEA1 positive endosomes present in WT cells as it induced large focal EEA1 accumulations, but these did not have the typical vacuolar appearance (Fig. S4 E). Also, in this case, this result may be due to a dominant negative effect as similar EEA1 positive clusters, rather than the typical vacuolar structures, were observed in WT cells over-expressing SHIP164 (Fig. S4 E). However, we confirmed the impact of SHIP164 deficiency on the presence of large EEA1 endosomes by SHIP164 siRNA-dependent knock-down in RPE-1 cells, thus ruling out off-target effects of CRISPR/Cas9 editing (Figs. S4, F and G; **, P < 0.01).

We conclude that loss of SHIP164 function may globally impact membrane traffic between the cell periphery and the Golgi complex, while over-expression of SHIP164 disrupts selectively the retrograde traffic of a subset of the vesicles with which SHIP164 is associated.

## Presence of ATG9A in vesicle clusters induced by SHIP164 overexpression

SHIP164 has a special structural similarity to ATG2 (Fig. 7 A) and has similar lipid transport properties as assessed by in vitro studies (Valverde et al., 2019; Noda, 2021; Osawa et al., 2019). ATG2 was found to function in cooperation with the scramblase ATG9 to allow equilibration of delivered lipids between bilayer leaflets (Guardia et al., 2020; Ghanbarpour et al., 2021; Matoba et al., 2020; Noda, 2021; Reinisch et al., 2021). In addition to its core role in autophagy, ATG9A also functions in other contexts (Claude-Taupin et al., 2021; Mailler et al., 2021; Campisi et al., 2022), prompting us to explore whether, in addition to ATG2, it might also partner SHIP164. In WT cells, ATG9A is present in the Golgi complex area and also as scattered dots throughout the cytoplasm (Fig. S5 A). Using the expression of ATG9A-mScarlet, only very few endogenous SHIP164^mNG foci were positive for ATG9A (Fig. 7 B). Moreover, we found no effect on the normal scattered cytoplasmic distribution of ATG9A in cells depleted of SHIP164 (Fig. S5, B and C). However, we did detect a striking overlap of overexpressed SHIP164 (SHIP164^mScarlet) with either exogenous ATG9A-GFP or endogenous ATG9A (detected by immunofluorescence; Fig. 7, C and D). As clusters of exogenous SHIP164 show specificity for vesicles based on cargoes—they are not enriched in TGN46 but are strikingly enriched in MPR (see above, Fig. S2, D–I)—this result raises the possibility of a functional relation between SHIP164 and ATG9. However, we excluded a role of SHIP164 in autophagy, as endogenous WIPI2 (Fig. 7 D) and LC3 (RFP-LC3; Fig. 7 E), two markers of

autophagosomes, were not localized on SHIP164 accumulations, strongly suggesting that these accumulations are not an autophagy-related structure. We also did not find any effect on the total autophagic flux in SHIP164 KO cells compared with WT cells, as assessed by the conversion of LC3 (Fig. 7 F), indicating that the function of SHIP164 is not required for autophagy. Thus, a functional partnership between SHIP164 and ATG9 needs to be further explored.

## UHRF1BP1, a SHIP164 paralog, is also localized in the endocytic pathway

UHRF1BP1 shares substantial primary sequence similarity (42% identity) and domain organization with SHIP164 (Otto et al., 2010; Fig. 8 A). Accordingly, phenotypes produced by expression of exogenous UHRF1BP1 (UHRF1BP1-Halo) revealed similarities to those produced by the expression of exogenous SHIP164. As in the case of SHIP164, UHRF1BP1-Halo localized to foci of varying size and fluorescence intensity, with the larger foci being concentrated in the central region of the cell (Fig. 8 B). In contrast to SHIP164 foci, which were primarily associated with early endosomes, UHRF1BP1 foci were predominantly associated with LAMP1-GFP–positive organelles (Fig. 8, C and D), in agreement with previous works, suggesting that UHRF1BP1 is a Rab7 effector (Fig. 8 D), rather than a Rab5 effector (Gillingham et al., 2019). However, we find, like SHIP164, UHRF1BP1 accumulations changed shape and could undergo fission (Fig. 8 E). We conclude that SHIP164 and UHRF1BP1 likely function in a similar manner but at different stations of the endocytic pathway.

## Discussion

Our study identifies SHIP164 as a bona fide member of the chorein motif protein family with a role in membrane traffic. The EM analyses of SHIP164 show that as in VPS13 and ATG2, the two well-established members of this family, the N-terminal chorein motif of SHIP164 caps an extended rod that harbors a cavity running along its length (Kumar et al., 2018; Valverde et al., 2019; Maeda et al., 2019; Li et al., 2020). Fold prediction algorithms indicate that SHIP164 features an extended β-sheet that resembles a taco shell, whose concave elongated surface is lined with hydrophobic residues, and so constitutes a groove that can solubilize multiple lipids (Jumper et al., 2021; Yang et al., 2020). In vitro lipid binding and transfer assays further support that SHIP164 might function as a lipid transfer protein. The SHIP164 paralog UHRF1BP1 is predicted to have a similar structure.

Our analysis of the subcellular localization of SHIP164 reveals an association with endocytic organelles, extending previous observations (Otto et al., 2010; Gillingham et al., 2019). This is supported by several pieces of evidence: (1) Endogenous SHIP164 is predominantly localized on a population of small peripheral vesicles that assemble in small clusters and contain in their membranes endosomal cargo proteins, such as MPR (Lin et al., 2003). (2) Over-expressed SHIP164 drastically enlarges the size of these clusters, which become, in many cases, tightly associated with vacuoles positive for markers of early endosomal

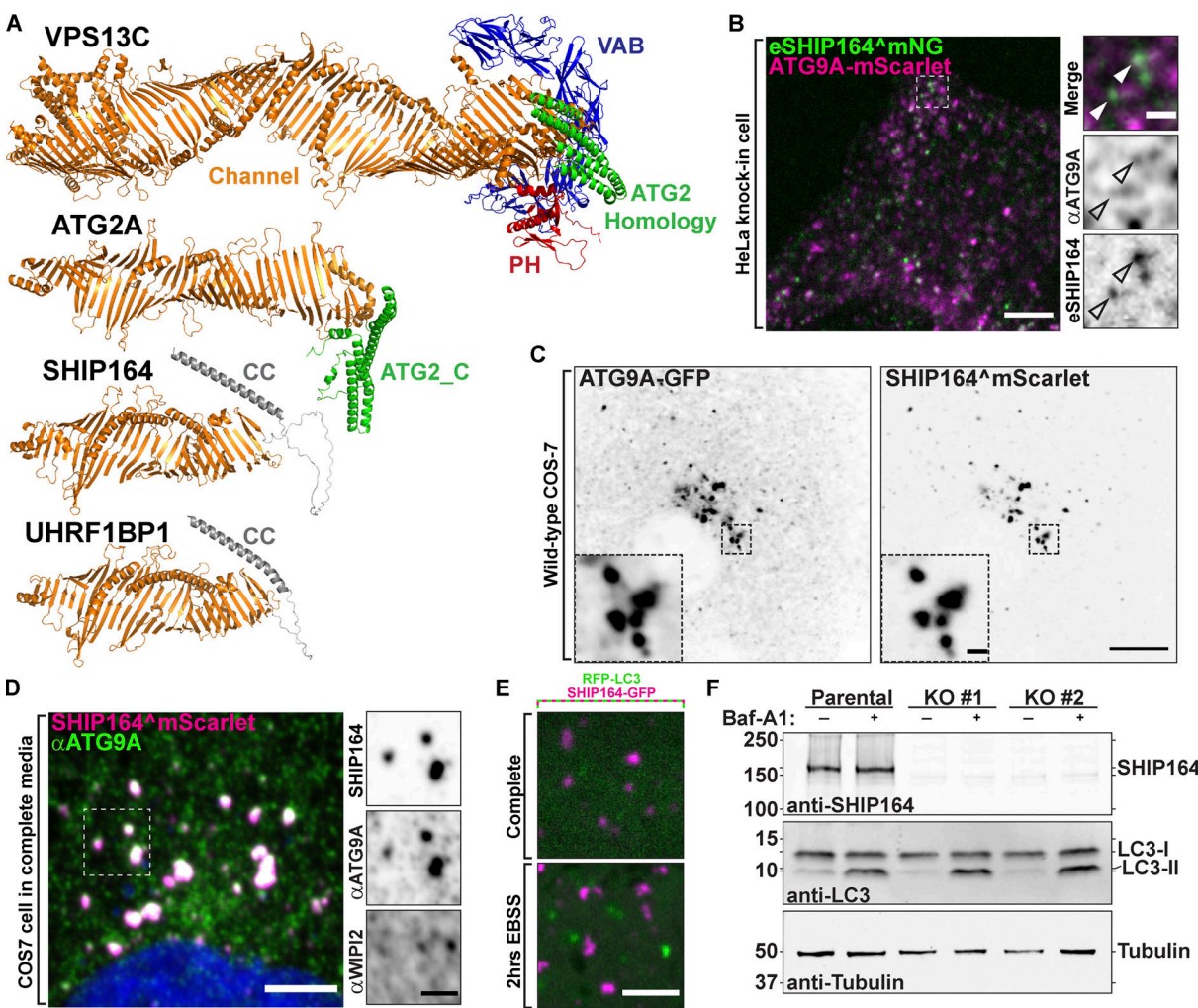

Figure 7. **Selective properties of SHIP164 vesicle clusters. (A)** Structures of the chorein motif proteins SHIP164, its paralog UHRF1BP1, ATG2A, and VPS13C at the same scale as predicted by the fold prediction algorithm AlphaFold. **(B)** Live fluorescence image of an endogenously tagged SHIP164^mNG (green) HeLa cell expressing exogenous ATG9A-mScarlet (magenta). Scale bar, 5 µm. High magnification of the indicated regions is shown at right. Arrowheads indicate potential overlapping fluorescence. Scale bar, 1 µm. **(C)** Live fluorescence (inverted grays) image of a COS-7 cell expressing exogenous ATG9A-GFP (left) and SHIP164^mScarlet expression (right). Scale bar, 10 µm. High-magnification scale bar, 1 µm. **(D)** Fluorescence image of a fixed COS-7 cell expressing exogenous SHIP164^mScarlet (magenta) and immunolabeled with antibodies against ATG9A (green) and WIPI2 (shown in high magnification). Scale bar, 10 µm. High-magnification scale bar, 1 µm. **(E)** Live image of the cytoplasm of a COS-7 cell expressing exogenous SHIP164-GFP (magenta) and the autophagosome marker RFP-LC3 (green) in either complete medium (top) or starvation conditions (bottom). Note the lack of colocalization between these proteins. Scale bar, 4 µm. **(F)** Western blot (in kD) of RPE1 cells for SHIP164, LC3, and for tubulin as a loading control, either treated with Bafilomycin A1 or DMSO demonstrating normal autophagic flux in SHIP164 KO cells. Source data are available for this figure: SourceData F7.

sub-compartments and are found also in deep regions of the cell. (3) The interactors of SHIP164 that we have identified or validated are endocytic factors or proteins that mediate retrograde organelle traffic from the cell periphery to the centrosomal/Golgi complex area. One of them is Rab5, an early endosomal Rab (Wandinger-Ness and Zerial, 2014). Another one is Stx6, a component of SNARE complexes, which mediates the fusion of vesicles with recycling endosomes and with the Golgi complex and whose action is facilitated by the GARP and EARP tethering complexes (Simonsen et al., 1999; Bock et al., 1996; Ganley et al., 2008). Interestingly, binding of SHIP164 to the Habc domain of Stx6 is mediated by the same motif used by the GARP and EARP subunit VPS51, also a Stx6 interactor (Pérez-Victoria and Bonifacino, 2009; Abascal-Palacios et al., 2013). While this finding

implies a competitive binding to Stx6, it also suggests an interplay of SHIP164 with GARP, further linking SHIP164 to retrograde traffic. A third interactor is the light chain of dynein (DYNLL1/2), the microtubule motor that mediates retrograde traffic (Rapali et al., 2011b). The fourth is Rab45, an unconventional Rab that anchors endocytic vesicles to dynein, creating an additional link to dynein and retrograde traffic (Wang et al., 2019; Shintani et al., 2007). Notably, however, SHIP164, when expressed at endogenous levels, is predominantly associated with peripheral vesicles, suggesting a regulated interaction with vesicles that are lost and replaced by other interactors as they travel to deeper destinations in the cell.

A role of SHIP164 in retrograde traffic is further supported by SHIP164 loss-of-function and gain-of-function studies. In

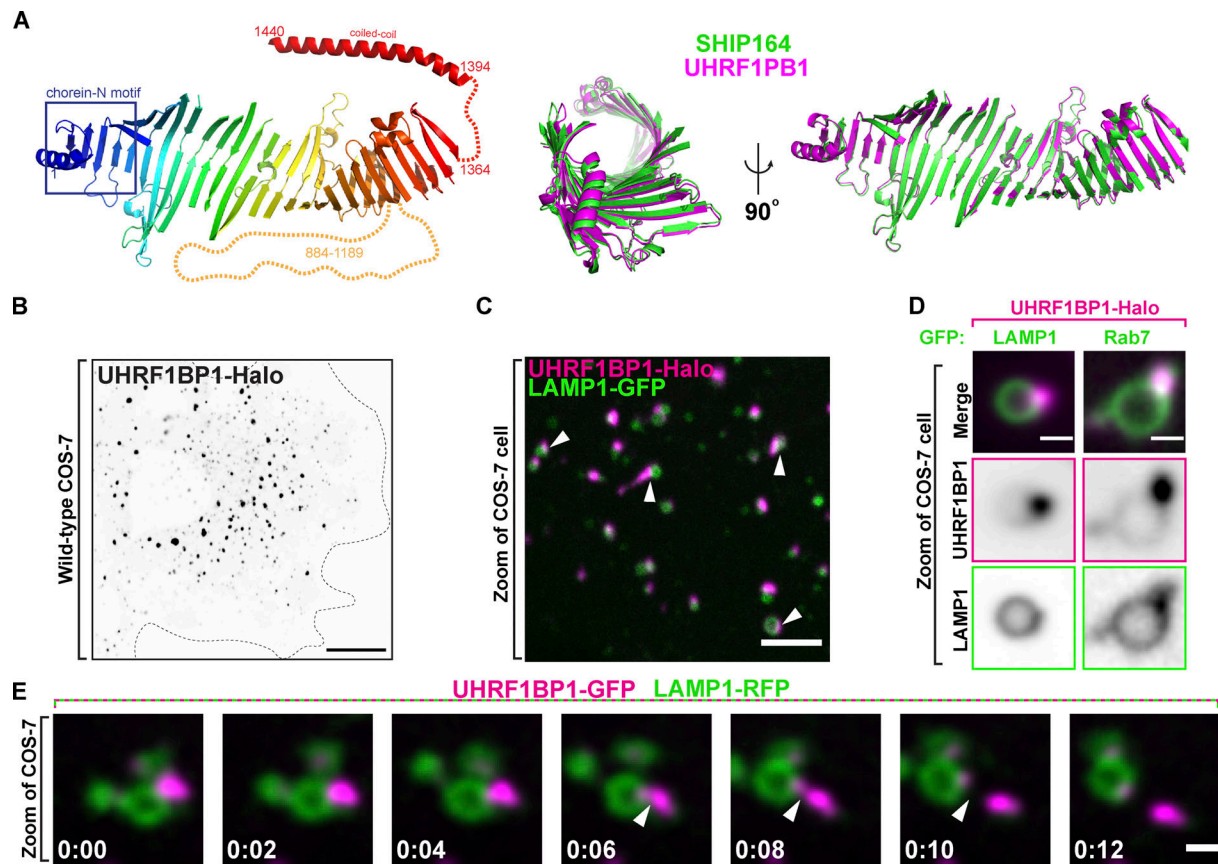

Figure 8. **The SHIP164 paralog UHRF1BP1 localizes on later endosomal compartments relative to SHIP164. (A)** Left: The strucutre of UHRF1BP1 as predicted by the fold prediction algorithm AlphaFold. Middle and Right: Alignment of the core channel structures of SHIP164 (green) and UHRF1BP1 (purple) in two different orientations showing structural similarity. **(B)** Live fluorescence (inverted grays) image of a COS-7 cell expressing exogenous UHRF1BP1-Halo. Scale bar, 20 µm. **(C)** Live image of the cytoplasm of a COS-7 cell expressing exogenous UHRF1BP1-Halo (magenta) and the endolysosomal marker LAMP1-GFP (green). Arrowheads indicate SHIP164 accumulation juxtaposed to LAMP1-positive organelles. Scale bar, 5 µm. **(D)** High-magnification live fluorescence images of COS-7 cell expressing exogenous UHRF1BP1-Halo (magenta) and either LAMP1 or Rab7 (green). The individual channels are shown as inverted grays. Scale bar, 2 µm. **(E)** Time-series of live fluorescence images of exogenous UHRF1BP1-GFP (magenta) and LAMP1-RFP (green). Arrowheads point to a UHRF1BP1 accumulation undergoing fission. Time, seconds. Scale bar, 1 µm.

SHIP164 KO cells, a larger pool of proteins that shuttle between the cell periphery and the Golgi complex (MPR, sortilin, and TGN46) is localized on vesicles scattered throughout the cytoplasm relative to the major pool localized in the Golgi complex area, suggesting a defect in transport/docking or fusion with the Golgi complex. Conversely, SHIP164 over-expression (gain-of-function) results in large vesicle clusters positive for MPR and sortilin, but not for TGN46, indicating that loss-of-function and gain-of-function effects are not precise opposites. A potential scenario is that the accumulation of sortilin and MPR vesicles results from a block in their progression to the next traffic station produced by excess SHIP164. Such dominant negative effects could be the result of sequestration of or competition with critical factor(s) by over-expressed SHIP164 or by alteration of membrane lipid homeostasis.

A key question is how the structure and molecular properties of SHIP164, which predict a lipid transport function at intracellular membrane contact sites, similar to the proposed function of VPS13 and ATG2, relate to the observed localization and loss-of-function phenotypes of SHIP164. We consider the hypothesis that SHIP164 may function in the transport of lipids

from one membrane, where the chorein domain may extract lipids—for example, the ER—to other membranes (Leonzino et al., 2021) and more specifically to the small vesicles enriched with endogenous SHIP164. This transport may deliver specific lipids or simply be responsible for membrane expansion and for controlling the protein–lipid ratio of a membrane as it moves to deeper positions in the cell. However, the localization of SHIP164 that we have observed is not at contacts of two types of membranous organelles, as predicted by this model. While some SHIP164 foci detectable by fluorescence appear to be localized at the interface of the ER and endosomes, such foci are represented by vesicle clusters and not by SHIP164 tethers between the ER and endosomes. Moreover, until now we have not identified the binding of SHIP164 to any ER protein. SHIP164 does not have an FFAT or FFAT-like motif that, as in the case of VPS13 (Kumar et al., 2018; Guillén-Samander et al., 2021), could account for binding to the ER protein VAP (Murphy and Levine, 2016; Cabukusta et al., 2020). In spite of these considerations, binding of SHIP164 to the ER cannot be excluded. Binding to the ER of SHIP164, tagged at the endogenous locus, is difficult to analyze given the small size of SHIP164-associated vesicles.

SHIP164 over-expression may overwhelm a binding site on the ER, unless such binding site was also over-expressed. One should also consider that an interaction with the ER of SHIP164-positive transport vesicles may be regulated and only transient.

Of potential interest in the context of protein-mediated lipid transport is the accumulation of SHIP64 with ATG9A in SHIP164 over-expressing cells as ATG9A is a scramblase (Gómez-Sánchez et al., 2018; Guardia et al., 2020; Matoba et al., 2020; Ghanbarpour et al., 2021). Scramblases are expected to be functional partners of VPS13-ATG2 family members, as bulk transfer of lipids between cytosolic leaflets of adjacent bilayers requires scramblases to allow equilibration with the noncytosolic leaflets (Guardia et al., 2020; Matoba et al., 2020; Ghanbarpour et al., 2021). While ATG9A was discovered as a core component of the autophagy machinery, it is plausible that it may function in other contexts, as also recently reported (Claude-Taupin et al., 2021; Mailler et al., 2021; Campisi et al., 2022). Interestingly, studies of ATG9A have consistently localized it to small vesicles often organized in clusters, similar to the SHIP164 vesicles described here (Davies et al., 2018; Mari et al., 2010).

Finally, SHIP164 was reported to be a Parkinson's disease candidate gene (Jansen et al., 2017), and loss-of-function mutations in VPS13C, another chorein motif lipid transport protein (Fig. 7 A), implicated in the endocytic/lysosome pathway are responsible for familial forms of Parkinson's disease (Lesage et al., 2016). It is therefore possible that SHIP164 may be part of a network of proteins that play an important role in neuronal health by controlling lipid transport and metabolism at the interface with membrane traffic.

## Materials and methods
### Antibodies and reagents
The list of antibodies, their working dilution, and the supplier for this study can be found in Table S1. All Halo-tag ligands used (JF549 and JF646) were a kind gift from Luke Lavis (Janelia Research Campus, Ashburn, VA). All SHIP164 (UHRF1BP1L) ORFs used in this study utilized a human codon-optimized sequence designed and purchased from Genscript (sequence available upon request). The following constructs were kind gifts: BFP-Rab5b and mito-BFP from G. Voeltz (University of Colorado-Boulder, Boulder, CO; Addgene plasmids #49147, #49151, respectively); Sec61b-GFP and RFP from T. Rapoport (Harvard University, Cambridge, MA); GFP-2xHrs$^{FYVE}$ from Harald Stenmark (University of Oslo, Oslo, Norway; Addgene plasmids #140047). Other plasmids encoding EEA1, WDFY2, APPL1, and 2xFYVE were previously generated in the De Camilli lab. All other ORFs used are listed in Table S2.

All EM reagents were purchased from Electron Microscopy Sciences.

Lipids were purchased from Avanti Polar Lipids: DOPC (850357), liver PE (840026), DGS-NTA (Ni; 790404), brain PI(4,5)P2 (840046), Rh-PE (810150), and NBD-PE (810145).

### Generation of plasmids
Codon optimized human SHIP164 generated by Genscript was amplified using PCR from the pUC57 plasmid and ligated into various mammalian and bacterial expression plasmids. Most constructs were generated with regular cloning protocols or through site-directed mutagenesis. The desired ORFs were amplified by PCR and inserted into plasmids through enzyme digestion and ligation. Some amplified ORFs were ligated using HiFi assembly (NEB). Details of primer sets, enzymes, techniques, and plasmids used for each construct can be found in Table S2.

For internal tagging of SHIP164, the codon-optimized human sequence was first amplified by PCR and cloned into p3xFLAG (Millipore Sigma plasmid E7658) between the EcoRI/KpnI restriction sites using HiFi assembly. From there, the ORF of mScarlet-I was amplified by PCR and ligated into the EcoRV site at residue 915 of SHIP164 using HiFi assembly. To generate the amino-terminal free version, SHIP164^mScarlet was amplified by PCR and ligated into pcDNA3.1 His-A (ThermoFisher plasmid V38520) between the HindIII/XhoI sites removing the His$_6$ tag. Unless stated otherwise, all mutations in this study were generated using site-directed mutagenesis. All ORFs were sequenced in their entirety after cloning and before use in any and all experiments.

For the MBP-SHIP164 fusion construct used in the in vitro analysis, sequences coding for residues 27–392 of *Escherichia coli* MBP and two additional alanines were fused to the N-terminus of SHIP164$_{\Delta901-1099}$ using Gibson Assembly and site-directed mutagenesis.

The tethered SHIP164 construct used to perform the in vitro lipid transfer assay was generated using Gibson Assembly, as previously described (Kumar et al., 2018), and consists of SHIP164$_{\Delta901-1099}$ followed by the unstructured region of human Esyt2(residues 649–689), then the pleckstrin homology domain of rat PLCΔ(residues 11–140), the unstructured region of Eyst2(residues 690–739), and a hexahistidine tag at the C-terminus.

Detailed protocol for the molecular cloning of SHIP164 plasmids for expression in mammalian cells is at: https://dx.doi.org/10.17504/protocols.io.8epv5z5kjv1b/v1.

### Protein expression and purification
For lipid transfer and cryo-EM studies, 3×FLAG-SHIP164$_{\Delta901-1099}$ constructs were transfected into Expi293F cells (Thermo Fisher Scientific) according to manufacturer instructions. Protein expression was enhanced by the addition of nonessential amino acids (Gibco) and valproic acid (3.5 mM final concentration) 16 h after transfection. Cells were harvested by centrifugation 65 h after transfection and either flash-frozen for storage or used immediately for protein purification. Cells were lysed by three freeze–thaw cycles in buffer (50 mM Hepes, pH 8.0, 500 mM NaCl, 1 mM TCEP, 5% glycerol) containing 1× complete EDTA-free protease inhibitor cocktail (Roche). Insoluble debris was removed by centrifugation at 15,000 rpm for 30 min, and clarified lysate was mixed with anti-FLAG M2 resin (Sigma-Aldrich) for 3 h while rotating at 4°C. Resin was then washed with 3 × 10 bed volumes of buffer and subsequently incubated for 18 h at 4°C in buffer supplemented with 1 mM ATP and 2 mM MgCl$_2$. Resin was washed again with three-bed volumes of buffer and then eluted in 5 × 0.4 bed volumes of buffer containing 0.25 mg/ml 3×FLAG peptide (APExBio). The protein was further purified by

gel filtration (Superdex 200 10/300 GL; GE Healthcare) and concentrated to ~1.5 μM (Amicon). E-Syt2 was expressed and purified as previously described (Schauder et al., 2014).

Detailed protocol for transfection and protein over-expression in Expi293 cells is at: https://dx.doi.org/10.17504/protocols.io.bp2l61x1kvqe/v1.

Detailed protocol for protein purification related to SHIP164 is at: https://dx.doi.org/10.17504/protocols.io.x54v9ypjqg3e/v1.

The MBP-SHIP164$_{1312-1464}$/DYNLL1 complex was recombinantly expressed in BL21 DE3 competent cells (Agilent Technologies; 230130) for purification and size exclusion chromatography analysis using the polycistronic pET Duet expression plasmid (Millipore Sigma plasmid #71146-3), details of which can be found in Table S2. BL21 cells transformed with the dual expression plasmid were grown at 37°C in super broth media (1 liter) until reaching an OD$_{600}$ between 2 and 3, after which the temperature was turned down to 18°C and cells were induced with 0.25 mM IPTG for 16 h with continuous shaking. Cells were pelleted in a floor centrifuge at 4,000 g for 15 min and then resuspended in lysis buffer (25 mM Tris, pH 7.5, 200 mM NaCl, 0.1% Tween-20) and frozen at –80°C until ready for purification. For purification, the pellet was thawed in a water bath and then sonicated until the desired lysis using a macro sonicator tip (VirTris VirSonic). Membrane and unsonicated material were pelleted in a floor ultra-centrifuge at 24,000 g for 20 min, and the supernatant was collected and incubated with 250 μl of amylose resin for 1 h rotating at 4°C. The resin was then washed three times with 40× bed volumes of ice-cold wash buffer (25 mM Tris, pH 7.5, 200 mM NaCl, 1 mM BME). Two elutions were performed from the resin using 2× bed volume of elution buffer (25 mM Tris, pH 7.5, 150 mM NaCl, 10 mM maltose, 1 mM BME). The eluate was collected and run over a Superdex S200 column using a GE AKTA pure system, and individual fractions (1 ml) were collected for analysis by SDS-PAGE and Coomassie staining.

## Lipidomics analysis of SHIP164 by mass spectrometry

Full-length human 3xFLAG-SHIP164 was expressed and purified as described above with the following modifications. Following immunoprecipitation, FLAG resin-bound SHIP164 was washed with 3 × 100 bed volumes of sonication buffer containing 1 mM BME and protease inhibitor cocktail for 20 min. After elution, the protein was buffer exchanged into 50 mM mass spectrometry grade ammonium acetate. The purified protein sample and 1 × 10$^7$ untransfected Expi293F cells were sent to Avanti Polar Lipids for analysis. Lipids were extracted in 2:1 (vol/vol) methanol:chloroform. The chloroform layer was dried, and lipids were reconstituted with internal standards for phosphatidylcholine, phosphatidylethanolamine, phosphatidylinisitol, phosphatidylserine, phosphatidic acid, phosphatidylglycerol, sphingomyelin, triacylglycerol, diacylglycerol, and cholesterol ester for quantification. As described by the company, the sample was injected into a reverse-phase C8 column with a gradient elution profile for the resolution required of each lipid class and detected by an AB Sciex 5500 tandem mass spectrometer. The molecular species of lipids were quantified based on internal standards and summed by lipid class.

## Native gel lipid-binding assay

Purified E-Syt2 and SHIP164$_{Δ901-1099}$ at indicated concentrations were mixed with methanol-dissolved NBD-PE (0.2 mM final concentration) in 20-μl total reaction volumes and incubated for 2 h on ice. Samples were loaded on 4–20% Mini-PROTEAN TGX Precast Gels and run for 3 h at 200 V. NBD fluorescence was visualized using an ImageQuant LAS4000 (GE Healthcare), and gels were stained with Coomassie Brilliant Blue R (Sigma-Aldrich) to visualize total protein. A standard curve was generated using the fluorescence values of E-Syt2, and the quantity of lipid bound to SHIP164 was analyzed using ImageJ (National Institutes of Health).

Detailed protocol for native gel lipid binding assay is at: https://dx.doi.org/10.17504/protocols.io.bp2l61ddkvqe/v1.

## FRET-based lipid transfer assay

Liposomes were prepared as previously described (Kumar et al., 2018). Briefly, lipids in chloroform were mixed in the indicated ratios and dried to thin films, then resuspended in buffer containing 50 mM Hepes, pH 8.0, 500 nM NaCl, 1 mM TCEP, and 5% glycerol to 1 mM total lipid concentration. Rehydrated lipids were then incubated at 37°C for 1 h before being subjected to 10 freeze–thaw cycles. Crude liposomes were freshly extruded through a polycarbonate filter with 100-nm pore size prior to transfer assays.

Lipid-transfer experiments were set up in 100-μl reaction volumes containing 0.125 μM protein, 25 μM donor liposomes (61% DOPC, 30% liver PE, 2% NBD-PE, 2% rhodamine-PE, and 5% DGS-NTA[Ni]), and 25 μM acceptor liposomes (65% DOPC, 30% liver PE, and 5% PI[4,5]P$_2$). Donor and acceptor liposomes were included in the reaction for 5 min prior to the addition of protein to establish a fluorescence baseline. The lipid transfer reaction was monitored via NBD fluorescence intensity detected every 1 min at 538 nm upon excitation at 460 nm. Lipid-transfer assay was performed for 1 h at 30°C in a 96-well plate (Nunc) using a Synergy H1 Plate Reader (BioTek).

A tether-only construct consisting of an N-terminal hexahistidine-tagged plexstrin homology domain from rat PLCΔ(11–140) was expressed and purified as previously described (Bian et al., 2018). The lipid transfer assay using this construct was performed using the same protein:lipid ratio.

A dithionite assay, to rule out the possibility that the NBD fluorescence increase was due to SHIP164-mediated liposome fusion, was performed in the same way as the transfer assay, except for the final addition of 5 μl freshly prepared dithionite buffer (100 mM dithionite [Sigma-Aldrich] in 50 mM Tris, pH 10) after fluorescence increase observed during the transfer assay had reached its maximum. NBD emission was then monitored for an additional 30 min.

Detailed protocol of the lipid transfer assay is at: https://dx.doi.org/10.17504/protocols.io.8epv59x9jg1b/v1.

## Cryo-EM sample preparation, data collection, and image processing

Purified SHIP164$_{Δ901-1099}$ was supplemented with 0.05% n-octyl-β-D-glucoside immediately before cryo-EM samples were prepared. Quantifoil R1.2/1.3 300 mesh copper grids were glow-

discharged for 30 s at 22 mA, then 4 µl of sample was applied. The grids were blotted with a single blotting paper for 5.5 s at a blot force of –2 in 90% humidity at 4°C before being plunge-frozen in liquid ethane using an FEI Vitrobot Mark IV (Thermo Fisher Scientific).

Data collection was performed on a Titan Krios G2 transmission electron microscope (Thermo Fisher Scientific) at 300 kV with a K3 summit direct detection camera (Gatan). SerialEM was used to collect 5,381 movies at a nominal magnification of 81,000× in super-resolution mode with a magnified pixel size of 1.068 Å on the specimen level (counting mode). Movies were dose-fractionated into 43 frames of 0.08 s each at a dose rate of 16.9 electrons/Å$^2$/s for a total dose of 51 e$^-$/Å$^2$. The defocus range for the sample was between –1.4 and –2.4 µm.

Image processing was carried out using RELION-3.1 (Zivanov et al., 2020). The 5,381 micrographs were motion-corrected and dose-weighted in MotionCor2 (Zheng et al., 2017) with a binning factor of 2 and divided into 5 × 5 patches. The contrast transfer function was calculated with CTFFIND-4.1 (Rohou and Grigorieff, 2015). Reference-free 2D classification of 1,044 manually picked particles was done, and the three best classes that showed well-defined particles were used as references for autopicking, which yielded a dataset of 3.36 million particles. Multiple rounds of 2D classification were carried out to remove ice contamination and bad particles. The subset of the most homogeneous 254,169 particles was used to generate a 3D initial model using stochastic gradient descent with C1 symmetry as implemented in RELION, and 3D classification further isolated 86,720 particles. This 3D class underwent auto-refinement and postprocessing for a final reconstruction at a resolution of 8.3 Å according to the Fourier shell correlation (FSC) = 0.143 criterion.

Detailed protocol for cryo-EM sample preparation is at: https://dx.doi.org/10.17504/protocols.io.q26g74mm3gwz/v1.

## CLEM

For TEM CLEM, COS-7 cells were plated on 35-mm MatTek dish (P35G-1.5-14-CGRD) and transfected as described above with SHIP164^mScarlet, GFP-WDFY2, mito-BFP, and Ras$^{G12V}$. Cells were prefixed in 4% PFA in Live Cell Imaging Buffer (see above), then washed before fluorescence light microscopy imaging. Regions of interest were selected and their coordinates on the dish were identified using phase contrast. Cells were further fixed with 2.5% glutaraldehyde in 0.1 M sodium cacodylate buffer, postfixed in 2% OsO$_4$ and 1.5% K$_4$Fe(CN)$_6$ (Sigma-Aldrich) in 0.1 M sodium cacodylate buffer, en bloc stained with 2% aqueous uranyl acetate, dehydrated, and embedded in Embed 812. Cells of interest were relocated based on the pre-recorded coordinates. Ultrathin sections (50–60 nm) were observed in a Talos L 120C TEM microscope at 80 kV, and the images were taken with Velox software and a 4k × 4K Ceta CMOS Camera (Thermo Fisher Scientific). For FIB-SEM CLEM, mito-BFP was expressed in endogenously tagged eSHIP164^mNG HeLa cells processed as above except the dehydration was implemented at low temperatures gradually decreased from 0 to –50°C. Epon blocks were glued onto the SEM sample mounting aluminum, and platinum en bloc coating on the sample surface was carried out with the sputter coater (Ted Pella, Inc.). Samples were FIB-

SEM imaged in a Crossbeam 550 FIB-SEM workstation operating under SmartSEM (Carl Zeiss Microscopy GmbH) and Atlas engine 5 (Fibics incorporated). The imaging resolution was set at 5 nm/pixel in the x, y axis with milling being performed at 2.5 nm/step along the z axis (binned down by 2–5 nm when images were exported) to achieve an isotropic resolution of 5 nm voxel. Images were aligned and exported with Atlas 5 (Fibics Incorporated), further processed, and analyzed with DragonFly Pro software (Object Research Systems [ORS] Inc.). Except noted, all reagents were from Electron Microscopy Sciences.

Detailed protocol for 2D TEM CLEM is at: https://dx.doi.org/10.17504/protocols.io.261gend2jg47/v1.

Detailed protocol for 3D FIB-SEM CLEM is at: https://dx.doi.org/10.17504/protocols.io.6qpvr63rzvmk/v1.

## Cell culture and transfections

hTERT-RPE1 cells were a kind gift of A. Audhya (University of Wisconsin, Madison, Madison, WI). HeLa and COS-7 cells were obtained from American Type Culture Collection. All mammalian cells were maintained at 37°C in humidified atmosphere at 5% CO$_2$ unless noted otherwise. HeLa and COS-7 cells were grown in DMEM and RPE1 cells in DMEM/F12 medium (Thermo Fisher Scientific) supplemented with 10% FBS, 100 U/ml penicillin, 100 mg/ml streptomycin, and 2 mM glutamax (Thermo Fisher Scientific). Expi293F were grown to manufacturer's specifications in Expi293 expression medium with constant shaking. All cell lines were routinely tested and always resulted free from mycoplasma contamination.

Transient transfections were carried out on cells that were seeded at last 8 h prior. All transfections of plasmids used FugeneHD (Promega) according to manufacturer's specifications for 16–24 h in complete media without antibiotics. Expi293 transfections were carried out to manufacturer's specifications and reagents. siRNAs (IDT) for knock-down experiments against SHIP164 (design ID: hs.Ri.UHRF1BP1L.13.2) were transfected using Lipofectamine RNAiMAX (Thermo Fisher Scientific) for 48 h before fixation for immunocytochemistry experiments or collected for immunoblotting (see below).

Detailed protocol for cell culture, transfection, immunocytochemistry, and imaging: https://dx.doi.org/10.17504/protocols.io.eq2lyp55mlx9/v1.

## Immunoblotting and imaging procedure

All cell samples analyzed by immunoblotting (RPE1 and HeLa) were scraped from plates and harvested by centrifugation (500 g for 5 min). The pellet was washed with ice-cold PBS and centrifuged again in a 1.7-ml Eppendorf tube. The cells pellet was resuspended in lysis buffer (20 mM Tris-HCl, pH 7.5, 150 mM NaCl, 1 mM EDTA) containing protease inhibitor cocktail (Roche) and lysed using mechanical disruption (Isobiotec Cell Homogenizer). The lysate was clarified by centrifugation (17,000 g for 10 min) and then mixed with 3× SDS sample buffer (Cold Spring Harbor) to 1× concentration and then heated to 95°C for 5 min. A small portion of lysate was reserved for quantification of protein concentration by Bradford. 15–25 µg of protein samples were separated by electrophoresis on a 4–20% Mini-PROTEAN TGX gel and then subjected to standard Western

blot transfer and procedures. Blots were imaged using the Odyssey imaging system (Licor) using manufacturer's protocols. All primary antibodies used in this study are listed in Table S1.

## Live cell imaging and immunofluorescence

For all live-cell microscopy, cells were seeded on glass-bottom mat-tek dishes (MATtek corporation), 5,500/cm² in complete media. Transfections were carried out as described above. Spinning-disk confocal imaging was performed 16–24 h after transfection using an Andor Dragon Fly 200 (Oxford Instruments) inverted microscope equipped with a Zyla cMOS 5.5 camera and controlled by Fusion (Oxford Instruments) software. Laser lines used: DAPI, 440 nm; GFP, 488; RFP, 561; Cy5, 647. Images were acquired with a PlanApo objective (60 × 1.45-NA). During imaging, cells were maintained in Live Cell Imaging buffer (Life Technologies) in a cage incubator (Okolab) with humidified atmosphere at 37°C.

Halo-tag ligands were used at a final concentration of 200 nM. Cells were incubated with indicated dye for 45 min in complete media, rinsed three times with complete media, and then incubated in complete media for 45 min before imaging. Immunofluorescent experiments were performed with cells grown on #1 glass coverslips. Cells were fixed with 4% PFA in fixation buffer (400 mM sucrose, 125 mM NaCal, 5 mM KCl, 1 mM NaH2PO4, 2 mM MgCl2, 5 mM glucose, 1 mM EGTA, 5 mM Pipes) for 10 min at 37°C, washed 3× with TBS (Tris-HCl, pH 7.5, 150 mM NaCl), permeabilized using 1× PBS containing 0.5% Triton for 10 min at RT, and then blocked using antibody dilution buffer (50 mM Tris, pH 7.5, 150 mM NaCl, 0.1% Triton, 4% BSA) for 1 h at 25°C. Coverslips were incubated with the indicated primary antibody diluted in the same buffer overnight at 4°C. Slides were washed with PBS containing 0.1% Triton to remove excess primary antibody and subsequently incubated with the indicated secondary antibody diluted in antibody dilution buffer for 45 min at RT in the dark prior to mounting.

## Generation of CRISPR edited cell lines

SHIP164 RPE1 knockout cells were generated by first identifying a gRNA directed to exon 2 of SHIP164 using a Cas9 target identification tool (CHOPCHOP). A single-stranded oligo (IDT) was synthesized containing the sense sequence and then cloned into PX458 (#48138; Addgene) and transfected into RPE1 cells. 48 h after transfection, cells were single-cell sorted using FACS Aria based on GFP expression into 96-well plates, and individual clones were grown out and subsequently examined by immunoblot analysis. Sanger sequencing was used to confirm the presence of mutations, which led to frameshifts and premature stop codons within the coding sequence of SHIP164. Two validated clones were used in subsequent studies.

Endogenous tagging of HeLa cells was carried out using the ORANGE system (Willems et al., 2020). Briefly, a gRNA targeted close to the desired site within the endogenous reading frame of the indicated target protein was identified and subsequently cloned into empty pORANGE (Addgene plasmid #131471) using HiFi assembly at the tandem BbsI site. The ORF of the indicated fluorescent protein (mNG or mScarlet) with linker regions to make an in-frame insertion was cloned into the HindIII/XhoI

sites of pORANE using standard restriction enzyme cloning. The finalized plasmid was transfected into WT HeLa cells using the transfection protocol described above. Cells were split once to keep confluency below 80%, and 5 d after transfection the cells were imaged using confocal microscopy. After visualizing cells positive for the correct fluorescent protein, cells were serially diluted and plated at 10 cells per well on a glass-bottom 96-well plate (Cellvis P96-1-N) and inspected for fluorescence when the wells were sub-confluent. Wells with positive fluorescence were expanded, serially diluted, and plated at one cell per well on a glass-bottom 96-well plate and inspected for fluorescence when the clones were sub-confluent. All gRNAs used in this study are listed in Table S2.

Detailed protocol for the Generation of knock-in and knock-out CRISPR/Cas9 editing in mammalian cells is at: https://dx.doi.org/10.17504/protocols.io.5jyl85x89l2w/v1.

## Mouse brain lysate preparation, affinity purification, and protein identification

Brains were collected from sacrificed WT C57BL/6J mice (Jackson Laboratory strain #000664). For each 1 g of whole brian material, 10 ml of buffer (25 mM Hepes, pH 7.4, 200 mM NaCl, 5% glycerol, protease inhibitors) was added. Mechanical lysis was performed using a glass dounce-homogenizer (15 strokes). Triton X-100 was added to 1%, and material was rotated at 4°C for 30 min. Material was centrifuged at 1,000 g to remove cell debris and the collected supernatant was centrifuged at 27,000 g for 20 min. The resulting supernatant was flash frozen and stored at –80°C until use.

Purified 3xFLAG-SHIP164 or FLAG-tagged control protein was immobilized on anti-FLAG resin as described above. Immobilized SHIP164 and control protein was incubated with mouse brain lysate for 16 h gently rotating at 4°C. Resin was spun down and the supernatant was removed before 2 × 10 bed volume washes using a wash buffer (25 mM Hepes, pH 7.4, 150 mM NaCl, 1 mM BME, protease inhibitors). Protein bound to resin was boiled in the same wash buffer, and eluate was transferred to a new Eppendorf tube. Protein in the eluate was precipitated using trichloroacetic acid on ice for 16 h, neutralized using Tris pH 8.5 buffer, and then the precipitate was pelleted in a table-top centrifuge. The precipitate was washed twice with ice-cold acetone and then dried using a speed-vac. Protein and peptide identification was achieved by shotgun tandem liquid chromatography with tandem mass spectrometry analysis. Analysis was performed using Scaffold 5 proteome software to identify promising identifications.

## Image processing, analysis, and statistics

Florescence images presented in this study are representative of cells imaged in at least three independent experiments and were processed with ImageJ software. The dimensions of some of the magnification insets or panels were enlarged using the *Scale* function on ImageJ.

Quantification of vesicle-to-Golgi ratio and peripheral structures: unprocessed images with two channels (GM130 for Golgi in one channel and the protein of interest in the other channel) were made into maximum intensity projections. A

mask of the Golgi complex was made using the GM130 signal and then used to subtract the Golgi signal from the signal of the entire cell protein of interest channel resulting in peripheral signal only. Outlines of individual cells were drawn, and the amount of peripheral signal was measured and then divided by the extracted Golgi signal for each cell to obtain the ratio. Where stated that the number of peripheral structures was counted, the subtracted peripheral image was made into a mask to identify structures of the indicated area and subsequently counted per cell.

Statistical analysis was performed with GraphPad Prism 7 software. Groups were compared using a two-tail unpaired Student $t$-test, and results were deemed significant when a P value was smaller than 0.05.

Detailed protocol for ectopic receptor and endosome size and count image analysis and quantification is at: https://dx.doi.org/10.17504/protocols.io.q26g78jyklwz/v1.

### Online supplemental material
Fig. S1 contains more on the purification and structural analysis of SHIP164. Fig. S2 contains more of the localization of exogenous SHIP164 on endosomal compartments. Fig. S3 contains more details on the relationship between SHIP164 and Rab45. Fig. S4 contains more details on the connection between the loss of SHIP164 and the endosomal system. Fig. S5 contains more on the connection between SHIP164 and ATG9A. Table S1 is the list of antibodies used in this study. Table S2 contains a list of all oligonucleotides used in this study. Data S1 shows lipidomics data for Fig. 1 B. Data S2 shows original proteomics data for Fig. 5 B.

## Data availability
All primary data associated with each figure has been deposited in the Zenodo repository and can be found using the following link: https://doi.org/10.5281/zenodo.6322951.

## Acknowledgments
We thank members of the De Camilli and Reinisch lab for critical reading of the manuscript. We also thank Shan Xu and Song Pang (Janelia Research Labs) for help and discussion with FIB-SEM experiments.

This work was in supported in part by grants from the National Institutes of Health (NIH; NS36251, DA18343, and DK45735), the Kavli Institute for Neuroscience, and the Parkinson's Foundation to P. De Camilli; the NIH (R35GM131715) to K.M. Reinisch; the NIH (F32NS108448) to M.G. Hanna; and the NIH (T32GM008283) to P.H. Suen. This research was also funded in part by Aligning Science Across Parkinson's grant ASAP-000580 through the Michael J. Fox Foundation for Parkinson's Research to P. De Camilli and K.M. Reinisch.

P. De Camilli is a member of the Scientific Advisory Board of Casma Therapeutics. No conflict exists with this role. No other disclosures were reported.

Author contributions: M.G. Hanna, P.H. Suen, K.M. Reinisch, and P. De Camilli conceptualized the project. P.H. Suen performed structural analysis and lipid binding and transport experiments, Y. Wu performed CLEM experiments in collaboration with M.G. Hanna, and M.G. Hanna performed everything else. M.G. Hanna, P.H. Suen, Y. Wu, K.M. Reinisch, and P. De Camilli analyzed data. M.G. Hanna, P.H. Suen, K.M. Rwinisch, and P. De Camilli wrote the original draft of the manuscript, which was then reviewed and edited by all the authors.

Submitted: 7 November 2021

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

# Supplemental material

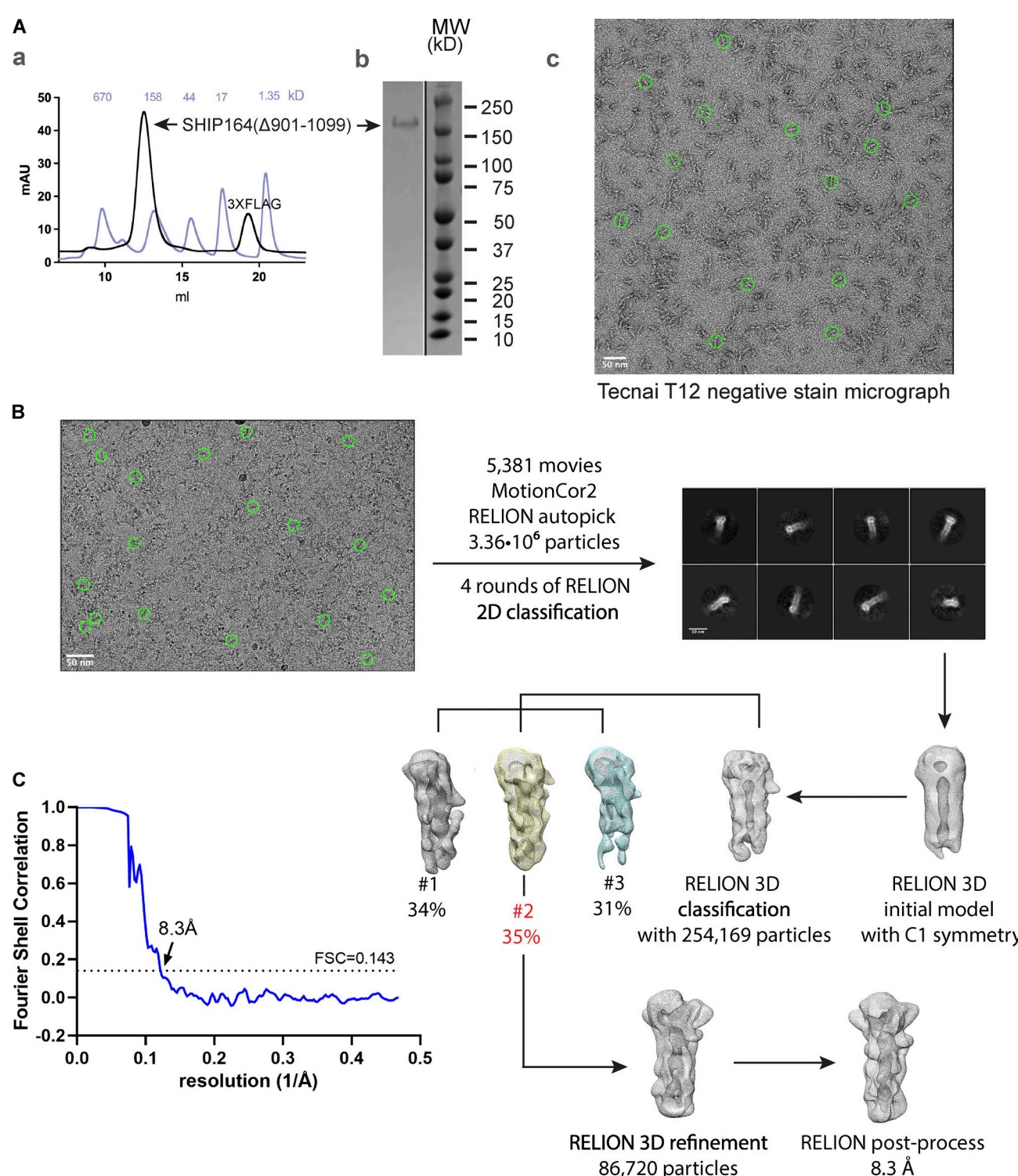

Figure S1. **In vitro characterization of SHIP164. (A)** (a) Following batch purification using anti-FLAG M2 resin, 3XFLAG-SHIP164$_{\Delta901-1099}$ was further purified by size exclusion chromatography on a Superdex-200 column. The gel filtration profile is shown (black), along with that for molecular weight standards (light blue). (b) Sample purity was examined by SDS-PAGE. (c) Representative negative stain electron micrograph of 3XFLAG-SHIP164$_{\Delta901-1099}$ (50 nM) using FEI Tecnai 12 microscope at 120 kV at a nominal magnification of 52,000×. Staining was with 2% uranyl acetate on carbon-coated copper grids that were glow-discharged for 30 s at 22 mA. Green circles represent particles manually picked with a 300 Å mask diameter in RELION-3.1. **(B)** Cryo-EM workflow. Micrographs, including the representative one shown, were collected using Titan Krios G2 transmission electron microscope (Thermo Fisher Scientific) at 300 kV equipped with a K3 summit direct detection camera (Gatan). Green circles represent particles manually picked for initial reference-free 2D classification prior to autopicking. **(C)** The processing workflow yielded a reconstruction with an estimated resolution of 8.3 Å according to the Fourier shell correlation = 0.143 criterion. Source data are available for this figure: SourceData FS1.

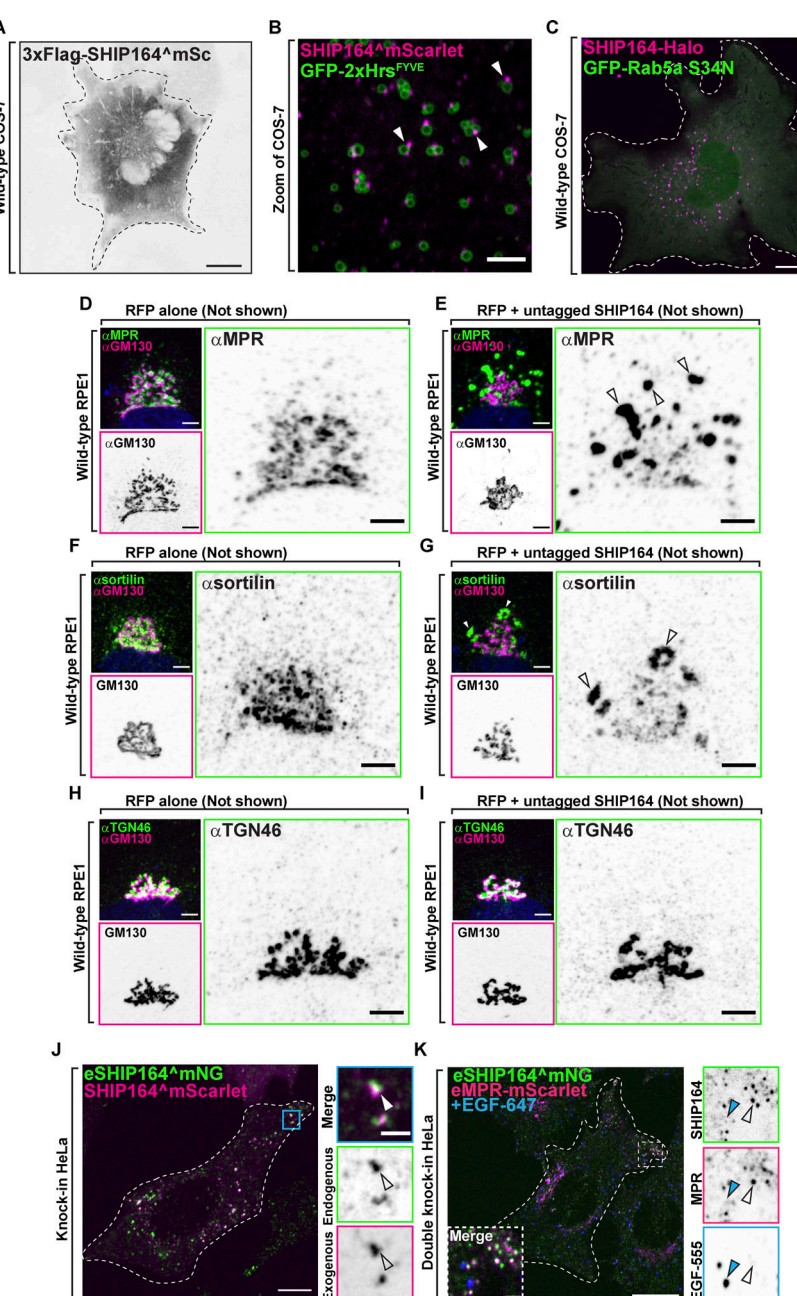

**Figure S2.  Exogenous SHIP164 forms foci. (A)** Live fluorescence (inverted grays) image of a COS-7 cell expressing exogenous 3xFlag-SHIP164^mScarlet. Scale bar, 20 µm. **(B)** Live image of the cytoplasm of a COS-7 cell expressing exogenous SHIP164^mScarlet (magenta) and the endosomal marker GFP-2xHrs^FYVE (green). Arrowheads indicate SHIP164 accumulation juxtaposed to endosomal membrane. Scale bar, 5 µm. **(C)** Live fluorescence image of a COS-7 cell expressing SHIP164-Halo (magenta) and the dominant negative GFP-Rab5 S34N mutant. Scale bar, 20 µm. **(D and E)** Fluorescence images of RPE-1 cells demonstrating normal Golgi complex localization of MPR in a cell expressing RFP alone (D) and its abnormal localization in a cell expressing both RFP and SHIP164 (E). Cells were immunolabeled with antibodies against MPR (green) and GM130 (magenta). Both the merge image and the single channels are shown. Arrowheads indicate ectopic accumulations of MPR in cells over-expressing SHIP164. Scale bar, 5 µm. **(F and G)** Fluorescence images of RPE-1 cells demonstrating normal Golgi complex localization of sortilin in a cell expressing RFP alone (F) and its abnormal localization in a cell expressing both RFP and SHIP164 (G). Cells were immunolabeled with antibodies against sortilin (green) and GM130 (magenta). Both the merge image and the single channels are shown. Arrowheads indicate ectopic accumulations of sortilin in cells over-expressing SHIP164. Scale bar, 5 µm. **(H and I)** Fluorescence images of RPE-1 cells demonstrating normal Golgi complex localization of TGN46 in a cell expressing RFP alone (H) and its unaffected localization in a cell expressing both RFP and SHIP164 (I). Cells were immunolabeled with antibodies against TGN46 (green) and GM130 (magenta). Both the merge image and the single channels are shown. Scale bar, 5 µm. **(J)** Merge fluorescence image of endogenous SHIP164^mNG (green) and exogenous SHIP164^mScarlet (magenta) in an edited HeLa knock-in cell to demonstrate overlap of the two fluorescence signals (arrowheads). Scale bar, 10 µm. High magnification of the indicated region is shown at right where the individual channels are shown as inverted grays. Scale bar, 2 µm. **(K)** Merge fluorescence image of endogenous MPR-mScarlet and endogenous SHIP164^mNG in a HeLa double-knock-in cell demonstrating partial colocalization of the two fusion proteins at the cell edge but not with pulsed EGF-647 (blue) foci. Scale bar, 10 µm. High magnification of the indicated region is shown at right and the individual channels are shown as inverted grays. Open arrowheads indicate overlapping fluorescence, and blue arrowheads an EGF-647–positive endocytic structure. Scale bar, 2 µm.

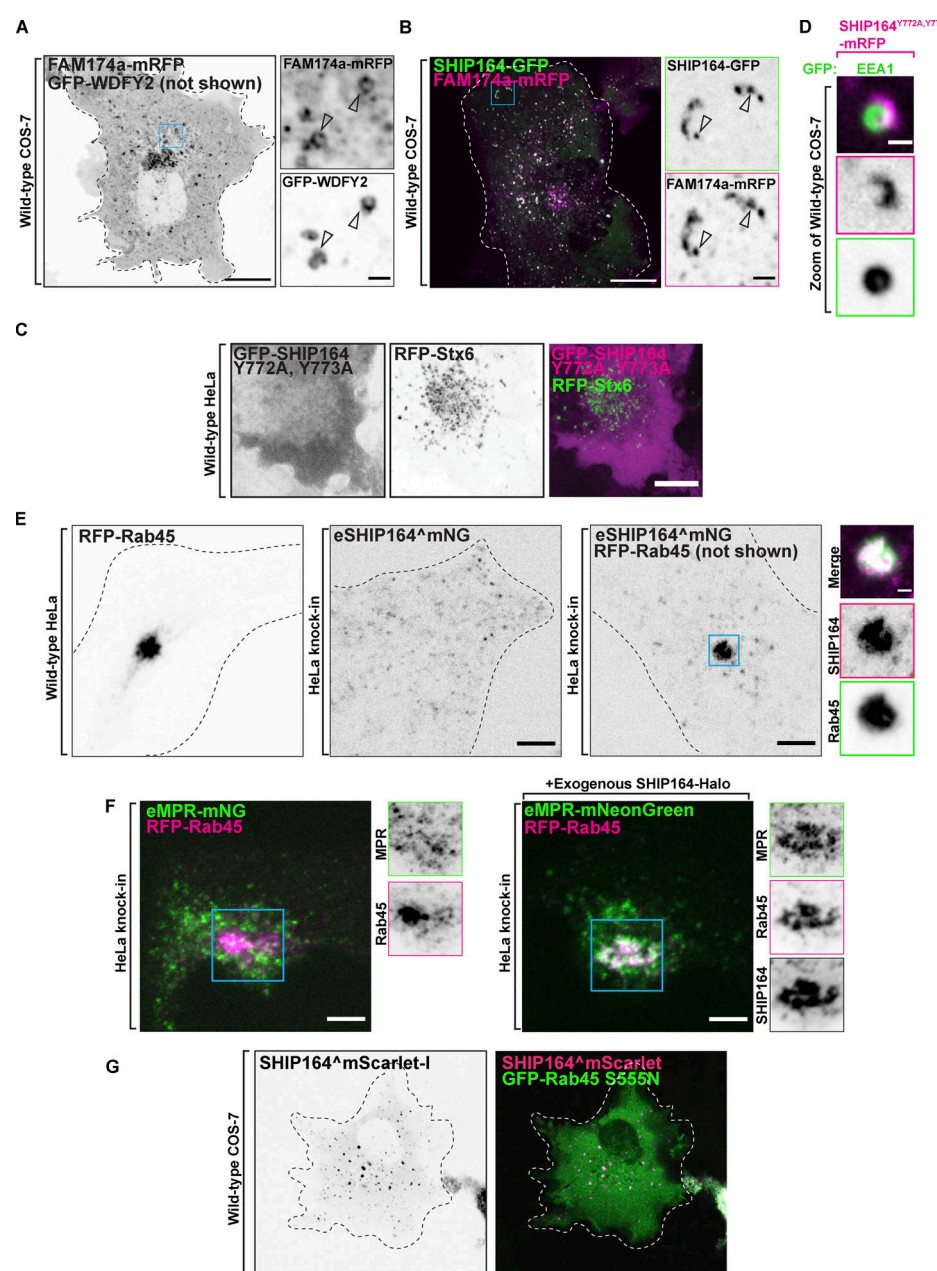

Figure S3. **SHIP164 localizes with endosomal proteins. (A)** Live fluorescence image of a COS-7 cell expressing FAM174a-mRFP (inverted grays) and GFP-WDFY2 (not shown). Scale bar, 20 μm. High magnification of the indicated regions is shown at right and the individual channels are shown as inverted grays. Arrowheads indicate overlapping fluorescence. Scale bar, 2 μm. **(B)** Live fluorescence image of a COS-7 cell demonstrating partial colocalization of exogenous SHIP164-GFP (green) FAM174a-mRFP (magenta) fluorescence in the cytoplasm. Scale bar, 10 μm. High magnification of the indicated regions is shown at right and the individual channels are shown as inverted grays. Arrowheads indicate overlapping fluorescence. Scale bar, 2 μm. **(C)** Live fluorescence images of a HeLa cell expressing exogenous GFP-SHIP164 Y772A, Y773A (magenta), and RFP-Stx6 (green) showing loss of Stx6 binding by this SHIP164 mutant. Individual channels are shown in inverted grays on the left. Scale bar, 10 μm. **(D)** High-magnification live fluorescence image of a COS-7 cell expressing exogenous SHIP164 Y772A, Y773A-mRFP, and the endosome marker EEA1 showing that this SHIP164 mutant still accumulates at hot spots. The individual channels are shown as inverted grays. Scale bar, 2 μm. **(E)** Live fluorescence images (inverted grays) of WT HeLa cell expressing RFP-Rab45, endogenous SHIP164^mNG in a HeLa knock-in cell, or HeLa knock-in cell expressing RFP-Rab45, as indicated. Scale bar, 10 μm. Note the massive recruitment of endogenous SHIP164 to the centrosomal region in the presence of Rab45. High magnification of the indicated regions is shown at right and the individual channels are shown as inverted grays. Scale bar, 2 μm. **(F)** Impact of the expresion of exogenous SHIP164 on the localization of endogenous MPR-mNG in cells also overexpressing Rab45. HeLa eMPR-mNG knock-in cells were transfected with Rab45 without (left) and with (right) the additional expression of SHIP164. Note that in the presence of SHIP164, Rab45 and MPR colocalize in a perinuclear spot. Left: Live fluorescence image of a HeLa knock-in cell demonstrating no change to the localization of endogenous MPR-mNG (green) upon expression of RFP-Rab45 (magenta). Right: Fluorescence image of a HeLa knock-in cell demonstrating colocalization of endogenous MPR-mNG and RFP-Rab45 upon the expression of SHIP164-Halo (not shown) in the centrosomal region. Scale bar, 10 μm. High magnification of the indicated regions is shown at right and the individual channels are shown as inverted grays. Scale bar, 2 μm. **(G)** Live fluorescence image of a COS-7 cell expressing SHIP164^mScarlet (magenta) and the dominant negative GFP-Rab45 S555N mutant (green) showing that this Rab45 mutant does not recruit SHIP164. Individual channel shown in inverted gray. Scale bar, 20 μm.

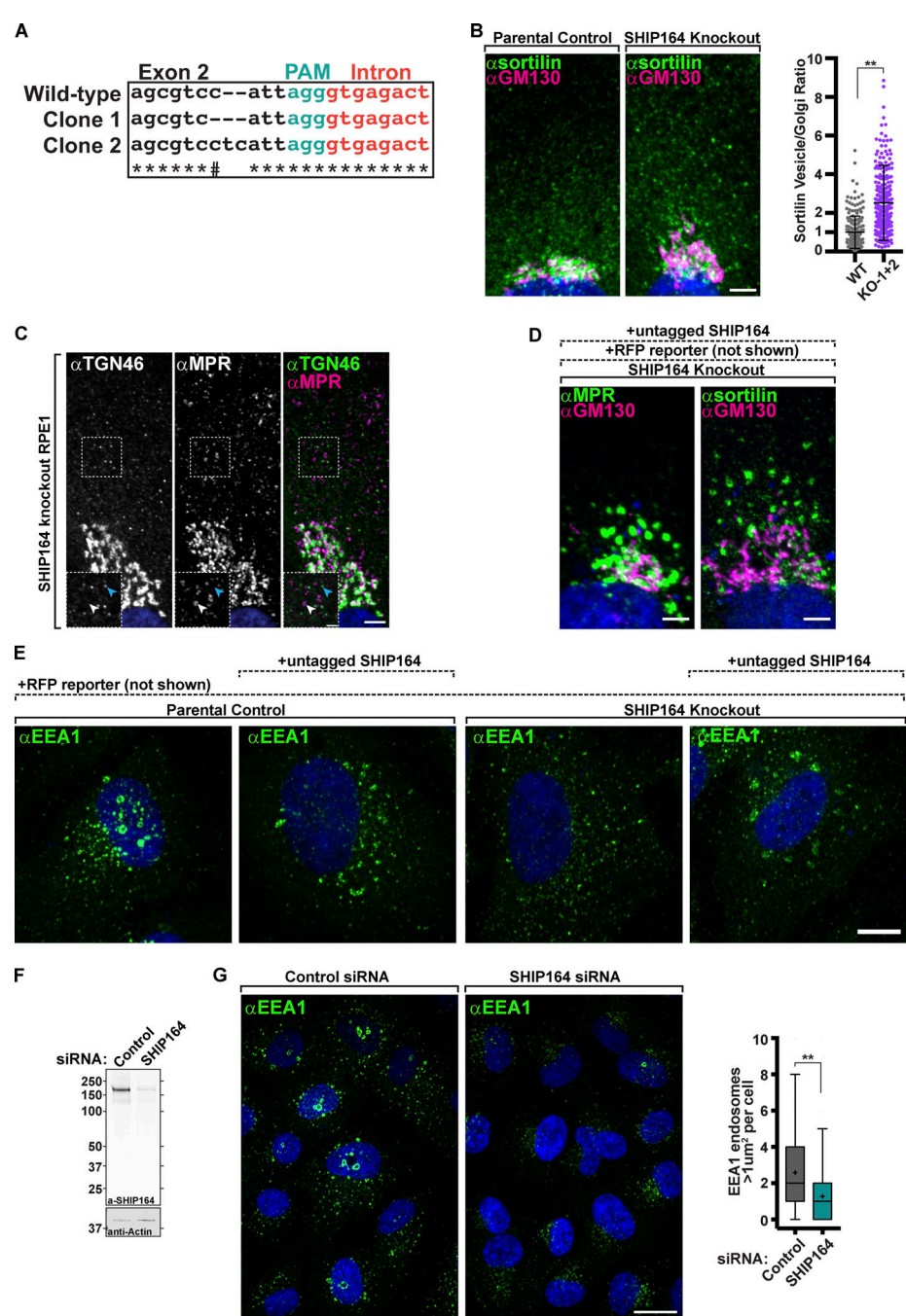

Figure S4. **Defects in retrograde traffic in SHIP164 loss of function cells. (A)** Sequencing results confirming editing of SHIP164 locus in RPE1 cells. PAM sequence (green) was directed to exon 2, and each clone was found to have a mutation starting four nucleotides upstream. # indicates nucleotide is the same between parental control and clone 2. **(B)** Left: Fluorescence images of a parental control (left) and SHIP164 KO cells (right) immunolabeled with antibodies against sortilin (green) and GM130 (magenta). Scale bar, 5 μm. Right: Quantification of the scattered cytoplasmic spots–to–Golgi complex ratio of sortilin signal per cell (circles) in control and SHIP164 KO cell clones. Middle line, mean; bars, SD. Data represents three biological replicates. **, P < 0.01. **(C)** Fluorescence images of a SHIP164 KO cell immunolabeled with antibodies against TGN46 (green) and MPR (magenta) demonstrating distinct post-Golgi vesicle populations. Scale bar, 5 μm. High magnifications of the indicated regions are shown as insets and the individual channels are shown as inverted grays. Arrowheads indicate divergent fluorescence. Scale, 2 μm. **(D)** Fluorescence images of a SHIP164 KO cells expressing both RFP and SHIP164 (not shown) immunolabeled with antibodies against the indicated proteins (green) and GM130 (magenta). Scale bar, 5 μm. **(E)** Fluorescence images of parental control or SHIP164 KO cells expressing either RFP alone (not shown) or RFP and exogenous untagged SHIP164 (not shown) immunolabeled with antibodies against the EEA1 (green) demonstrating that overexpression of SHIP164, even in KO cells, results in EEA1 accumulations not typically found in WT cells where SHIP164 is not over-expressed (far left). Scale bar, 10 μm. **(F)** Western blot (in kD) of RPE1 cells for SHIP164 and for GAPDH as a loading control, either treated with control or SHIP164 specific siRNAs. **(G)** Left: Fluorescence images of WT cells treated with control (left) or SHIP164 specific siRNAs (right) immunolabeled with antibodies against EEA1 (green). Scale bar, 20 μm. Right: Quantification of large EEA1 endosomes (>1 μm²) per cell in control and SHIP164 knock-down cell. Middle line, median; +, mean; bars, 10–90 percentiles. Data represents three biological replicates. **, P < 0.01. Source data are available for this figure: SourceData FS4.

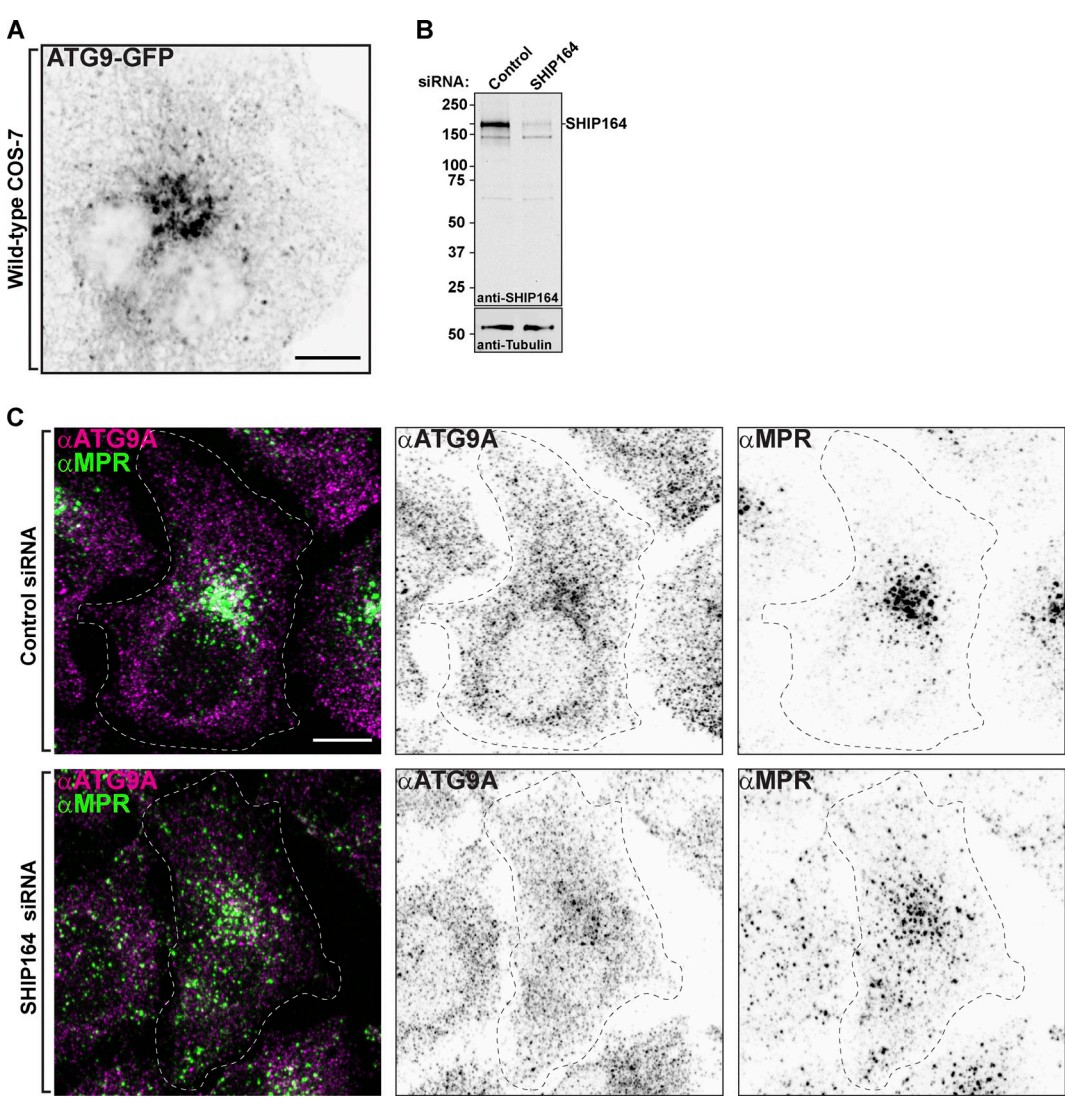

Figure S5. **Defects in retrograde traffic in SHIP164 depleted cells. (A)** Live fluorescence (inverted grays) image of a COS-7 cell expressing exogenous ATG9A-GFP. Scale bar, 10 μm. **(B)** Western blot (in kD) of HeLa cells for SHIP164 and for Tubulin as a loading control, either treated with control or SHIP164 specific siRNAs. **(C)** Fixed fluorescence images of WT HeLa cells treated with control (top) or SHIP164 specific siRNAs (bottom) immunolabeled with antibodies against ATG9A (magenta) and MPR (green). Scale bar, 10 μm. Source data are available for this figure: SourceData FS5.

Video 1. **High magnification of live fluorescence time series of COS-7 cell expressing exogenous SHIP164-mRFP (magenta) and the endosomal marker EEA1-GFP (green) demonstrating the dynamics of SHIP164 accumulations.** Frames captured every 2 s. 5 frames per second. Scale bar, 1 μm.

Video 2. **Zoom of live fluorescence time series of a double-knock-in HeLa cell demonstrating the colocalization of fluorescent signal from endogenous MPR-mScarlet (magenta; left) and endogenous SHIP164^mNG (green; right) puncta at the cell periphery.** Arrowheads indicate puncta positive for both mScarlet and mNG fluorescence. Frames captured every 2 s. 5 frames per second. Scale bar, 10 μm.

**Provided online are Table S1, Table S2, Data S1, and Data S2. Table S1 lists antibodies used in this study. Table S2 lists ORFs and primers used for cloning in this study. Data S1 shows lipidomics data for Fig. 1 B. Data S2 shows original proteomics data for Fig. 5 B.**

