## [Peer Review File · The Journal of Cell Biology]

SHIP164 is a Chorein Motif Lipid Transfer Protein that Controls Endosome-Golgi Membrane Traffic

Michael Hanna, Patreece Suen, Yumei Wu, Karin Reinisch, and Pietro De Camilli

Corresponding Author(s): Pietro De Camilli, Yale University and Karin Reinisch, Yale University Dept. of Cell Biology

Review Timeline:

Submission Date:	2021-11-07
Editorial Decision:	2022-01-05
Revision Received:	2022-03-07
Editorial Decision:	2022-03-29
Revision Received:	2022-04-07

Monitoring Editor: Mark von Zastrow

Scientific Editor: Andrea Marat

Transaction Report:

DOI: <https://doi.org/10.1083/jcb.202111018>

January 5, 2022

Re: JCB manuscript #202111018

Dr. Pietro V De Camilli
Yale University
Cell Biology and Neuroscience
295 Congress Avenue, BCMM 236
Lab BCMM 235-237
New Haven, CT 06519-1418

Dear Dr. De Camilli,

Thank you for submitting your manuscript entitled "SHIP164 is a Chorein Motif Lipid Transfer Protein that Controls Endosome-Golgi Membrane Traffic". The manuscript was assessed by expert reviewers, whose comments are appended to this letter. We invite you to submit a revision if you can address the reviewers' key concerns, as outlined here.

As you will see, both reviewers found your work to be very interesting, commented positively on the ambitious nature of the work, and noted that your study pulls together an impressive array of experimental approaches. However, both reviewers noted a number of key gaps in the data flow and controls. They also pointed out key aspects of your interpretations that do not appear adequately supported by data presented, or which would be considered overstatements based on the data presented.

We share the reviewers' high interest in your study and appreciate that it addresses a cutting-edge aspect of membrane cell biology. However, we also agree with them that there are some significant problems. Considering that your manuscript already has many strong aspects, and the overall topic addressed is fascinating, we invite you to prepare a revised manuscript addressing all of the reviewer critiques. We consider the reviewer critiques clear and constructive as written, but anticipate that some of them may be difficult to fully address in a realistic time frame. If this is the case, we would ask you to consider scaling back your revision to remove some claims that could be considered ancillary to (and possibly detract from) your study's already significant strengths. We would be happy to discuss any points that may not be feasible to address.

We ask you to pay particular attention to addressing critiques regarding relationships to autophagy. Both reviewers had concerns about your interpretations regarding SHIP164 colocalization with ATG9A, particularly as much of the data presented are based on protein overexpression. The reviewers made a number of suggestions for additional experiments to improve this part of the study and commented that leading with the overexpression data- while it is likely accurate with respect to the order of events as they were performed- is confusing to the outside reader because it requires walking back a number of interpretations with the knock-in results coming later. Thus we ask you to consider moving the overexpression data to supplementary, and focus the main data presentation to results obtained using native or near-native (endogenously tagged) expression. We also ask that you revise or scale back interpretations that rely on identifying autophagic membranes solely using LC3, per reviewer #2's comments. This would require claims made in the text to be appropriately scaled back, and perhaps converting a subset of them to speculations or future directions in the revised discussion.

We also ask that you pay close attention to statements regarding paralogues. You claim that UHRF1BP1 also localizes to endosome membranes and interacts / colocalizes with ATG9A. These claims appear to rely entirely on protein overexpression. We ask you to provide substantial additional support (at endogenous levels) and/or scale back, especially in light of previous evidence for localization to lysosomes rather than endosomes as noted by reviewer 1. We also suggest that you consider previous evidence (both reviewers) that ATG9 may function in the endocytic pathway separately from the proposed interactions with SHIP164 or UHRF1BP1 (such as with AP-1).

Finally, we ask that you include the full lipidomic and protein interaction data sets, and/or links to them in a repository. In the latter case, please specify (in supplement) the basis for selecting the hits that you focus on in the results section. We agree with the reviewers that, in its present form, the reported hits seem 'cherry-picked'.

GENERAL GUIDELINES:

Text limits: Character count for an Article is < 40,000, not including spaces. Count includes title page, abstract, introduction, results, discussion, acknowledgments, and figure legends. Count does not include materials and methods, references, tables, or supplemental legends.

Figures: Articles may have up to 10 main text figures. Figures must be prepared according to the policies outlined in our Instructions to Authors, under Data Presentation, <https://jcb.rupress.org/site/misc/ifora.xhtml>. All figures in accepted manuscripts will be screened prior to publication.

IMPORTANT: It is JCB policy that if requested, original data images must be made available. Failure to provide original images upon request will result in unavoidable delays in publication. Please ensure that you have access to all original microscopy and blot data images before submitting your revision.

Supplemental information: There are strict limits on the allowable amount of supplemental data. Articles may have up to 5 supplemental figures. Up to 10 supplemental videos or flash animations are allowed. A summary of all supplemental material should appear at the end of the Materials and methods section.

Please note that JCB now requires authors to submit Source Data used to generate figures containing gels and Western blots with all revised manuscripts. This Source Data consists of fully uncropped and unprocessed images for each gel/blot displayed in the main and supplemental figures. Since your paper includes cropped gel and/or blot images, please be sure to provide one Source Data file for each figure that contains gels and/or blots along with your revised manuscript files. File names for Source Data figures should be alphanumeric without any spaces or special characters (i.e., SourceDataF#, where F# refers to the associated main figure number or SourceDataFS# for those associated with Supplementary figures). The lanes of the gels/blots should be labeled as they are in the associated figure, the place where cropping was applied should be marked (with a box), and molecular weight/size standards should be labeled wherever possible. Source Data files will be made available to reviewers during evaluation of revised manuscripts and, if your paper is eventually published in JCB, the files will be directly linked to specific figures in the published article.

As you may know, the typical timeframe for revisions is three to four months. However, we at JCB realize that the implementation of social distancing and shelter in place measures that limit spread of COVID-19 also pose challenges to scientific researchers. Lab closures especially are preventing scientists from conducting experiments to further their research. Therefore, JCB has waived the revision time limit. We recommend that you reach out to the editors once your lab has reopened to decide on an appropriate time frame for resubmission. Please note that papers are generally considered through only one revision cycle, so any revised manuscript will likely be either accepted or rejected.

Thank you for this interesting contribution to Journal of Cell Biology. You can contact us at the journal office with any questions, cellbio@rockefeller.edu or call (212) 327-8588.

Sincerely,

Mark von Zastrow, MD, PhD
Monitoring Editor

Andrea L. Marat, PhD
Senior Scientific Editor

Journal of Cell Biology

Reviewer #1 (Comments to the Authors (Required)):

In this study, Hanna, Suen and colleagues examine the SHIP164 (UHRF1BP1L) protein which is a member of the Chorein lipid transfer protein family and is implicated in Parkinson's disease by a genome sequencing study. The authors perform convincing biochemical assays demonstrating that purified SHIP164 efficiently transfers lipids in vitro. Moreover, single particle cryo-electron microscopy experiments reveal that the SHIP164 structure resembles Chorein family members including VPS13 and ATG2, as recently predicted by AlphaFold. The authors also perform experiments to address the cellular function of SHIP164, suggesting roles in the trafficking of certain endosomal compartments from the cell periphery to perinuclear Golgi compartments and possibly autophagy. However, there are concerns that some of the functional tests rely on over-expression of the SHIP164

protein that results in indirect effects or artefacts. The authors take care to point out potential pitfalls in using the over-expressed protein, but then base some of their major conclusions on SHIP164 over-expression experiments. The purported association of SHIP164 with ATG9 vesicles is particularly egregious in this regard. Overall, this is an admirable study that incorporates biochemical, structural, and cell biological investigations, but additional tests are needed to substantiate some of the authors' conclusions regarding SHIP164 localization and function.

Major concerns:

1. The authors identify DYNLL1/2 as candidate SHIP164 binding partners. However, a functional significance of this association is lacking in the study. The authors should examine endogenously tagged SHIP164 (eSHIP164^{mNG}) upon depletion of DYNLL1/2. One may also expect that depletion of DYNLL1/2 would result in phenotypes similar to those observed in SHIP164 knockout (KO) cells, including differences in endosome size as well as localization of the mannose 6-phosphate receptor (M6PR), TGN46, and sortilin. Is this the case? In reciprocal experiments, is the distribution of DYNLL1/2 altered in SHIP164 KO cells?
2. The authors suggest that SHIP164 may be a Rab45 effector, but this is primarily based on experiments involving Rab45 over-expression. The authors should examine endogenously tagged SHIP164 (eSHIP164^{mNG}) upon depletion of Rab45. In addition, is M6PR distribution affected upon depletion of Rab45, similar to the SHIP164 KO cells?
3. The association of SHIP164 with ATG9 vesicles is tenuous. The authors found that over-expressed ATG9A-GFP is concentrated on SHIP164-positive foci upon SHIP164 over-expression (Figure 7B). Over-expression of SHIP164 along with Rab45 over-expression also increased the co-localization of these proteins with ATG9A (Figure 7C). However, since SHIP164 over-expression appears to induce "abnormal effects" upon endo-lysosomal compartments (Figure 3; lines 254-256), the over-expression data in Figure 7 does not 'prove' that ATG9A localizes to SHIP164-positive vesicles (lines 445-446). The authors should examine whether endogenously tagged SHIP164 (eSHIP164^{mNG}) co-localizes with ATG9A. The authors also observed that ATG9A is concentrated in the perinuclear Golgi complex and also present at compartments scattered throughout the cytoplasm (Figure 7A). Is the distribution of ATG9A altered in SHIP164 KO cells, similar to M6PR?
4. Likewise, the purported localization of the SHIP164 paralog UHRF1BP1 with ATG9A vesicles and LAMP1-positive lysosomes relies solely upon UHRF1BP1 over-expression experiments. Do the endogenous (or endogenously tagged) UHRF1BP1 and ATG9A proteins co-localize? The authors should also examine ATG9 vesicle and lysosomal (LAMP1) distribution in cells depleted of UHRF1BP1 in order to substantiate their conclusions.
5. A possible role of ATG9 vesicles functioning in endo-lysosomal trafficking along with SHIP164/UHRF1BP1L and/or UHRF1BP1 seems like an interesting and potentially novel concept (although there may be some evidence for this notion in the published literature, PMID: 29156099). The authors may want to directly test whether ATG9 isoforms regulate endosome size as well as localization of M6PR, TGN46, and sortilin, just as they did for SHIP164. It is curious that ATG9A/B were not implicated in previous studies on endosomal sorting pathways (distinct from their roles in autophagy & beside PMID: 29156099). However, redundancies could explain why this has been missed or overlooked. To further test this model, the authors may want to examine whether loss of ATG9A or ATG9B is additive with loss of SHIP164/UHRF1BP1L or UHRF1BP1.

Reviewer #2 (Comments to the Authors (Required)):

This is an important and beautiful study addressing some of the cutting edge issues in membrane biogenesis and bulk lipid transfer independent of vesicular fusion. The authors show that SHIP164, via its chorein motif, similar structurally to VPS13s and ATG2s, transfers bulk lipids. They show that it localizes to a cluster of small vesicles (presumed to be endocytic or of what origin?) with effects on endosomal-Golgi trafficking, mainly retrograde transport. Somewhat en passant/tangentially, an effect on autophagy was addressed since there is an unescapable similarity with ATG2s and apparent localization with ATG9A, etc. The hypothesis that SHIP164 is a lipid transport protein is supported by the results and is the main strength of this study, worth reporting. Should be of interest to many cell biologists.

However, it is necessary to strengthen the data behind the secondary objectives (past the lipid transfer) and improve the accessibility and interpretation of the results even concerning the primary objective. The absence/shortage of usable work with endogenous SHIP164 is a downside, but this plagues most of cell biology studies in general and is unavoidable. Similarly, in certain areas both the conceptual and technological approaches need to be improved/refined. In particular, the sections on autophagy need to be substantively and substantially revamped, and hierarchical issues regarding the roles of UHRF1BP1 and UHRF1BP1L need to be resolved. Nevertheless, this is an elegant and exciting study and this reviewer will be looking forward to seeing it in its revamped form should this happen.

Major comments:

1. A critical issue that needs to be addressed is the proposed role of SHIP164 in autophagy - knockout/mutational analysis is absolutely needed or else the whole connection (if at all) is circumstantial and substantially weakens the study.
2. The aspect of lipid transfer and comparative model with ATG2s and VPS13s is unescapable and well-presented/used in this

study. The point regarding ATG9A role outside of autophagy is exceptionally relevant and well taken. The authors may wish to elaborate a bit more. Because of the ATG2/ATG9A as comparators, it is critical to bring autophagy aspects of the study, currently severely curtailed to the current state of the field.

3. Autophagosomes form without LC3 (PMID: 34741801), with multiple studies in support of that (22275429, 27864321, 31353311, 27885029, 32320309). Thus, equating LC3 with autophagy (used here as a key argument) is no longer acceptable (albeit perpetuated in lower quality studies). Furthermore, LC3 is associated with various compartments including endosomal ones. The authors should bring their analysis of autophagy to a much more contemporary level and with appropriate rigor and provide experimental details that are at present entirely lacking (there is only narrative but no data).

4. Another key issue relates to UHRF1BP1 and UHRF1BP1L (SHIP164). The roles and relationships between these two paralogues and their inter-related roles in processes need to be disentangled, systematically addressed experimentally and hierarchically ordered within the pathways studied.

5. The last sentence of the abstract seems circular, rather than providing an expected clear conclusion. It reads as if the bulk lipid transfer plays a role in endosomal trafficking, whereas it is probably meant to say that endosomal trafficking plays a role in bulk lipid transfer. Even so, in this latter meaning, how does that help with the main premise, since the whole story is built on bulk lipid transfer independently of vesicular traffic, as stated? Perhaps the authors may want to recast this whole semantic exercise in the context of differentiating "vesicular fusion" rather than "vesicle traffic". A potentially good statement is found in the discussion, and would work well in the abstract (paraphrased/modified) : "controlling lipid transfer between membrane at the interface with vesicular trafficking" or a variant.

6. The shotgun lipidomics of lipids associated with the purified SHIP164 is very important and elegant. However, the summary representation in Fig. 1B is not satisfactory. Where are the primary data and if so can they be tabulated per exact lipid species (rather than the bland PC, PS, PI bar graphs? Were there any other lipids except phospholipids? How about cholesterol and sphingolipids? If yes, ratios to phospholipids (dis-enrichment) , if not, how is that interpreted?

7. It is confusing to first follow a lot of analyses with overexpressed SHIP164 fusions and then to learn that the authors conclude that much of that is an "over-expression defect" (line 309). Are all observations near the Golgi then an artifact? It would be good for authors to recast this work separating the "exploratory" stages, i.e. what belongs in notebooks and possibly can be all placed in a single supplementary figure illustrating some of those relationships that are per authors' own statement "artifactual", from "conclusive data". In other words, while some of the "historically" presented explorations are pretty it is disappointing to reach a conclusion that all we've been following is an artifact to a certain point. Please separate wheat from the chaff.

8. If ATG9A is found on "foci" resulting from overexpression, how is one to interpret this given the point re "overexpression defect" above?

9. Where are the complete MS data for SHIP164 interacting partners? Only cherrypicked dynein "data" are shown in Fig 5B.

10. It doesn't seem clear if the full-length construct of SHIP164 is prone to aggregate in all conditions and assays of the manuscript, especially in Figure 1. A related suggestion is to state in in figure legends what and when the authors used (e.g. the 3xFLAG-SHIP164 Δ 901-1099).

11. In the first paragraph of subheading `Localization of exogenous SHIP164...` (lines 172-176), the authors describe the localization of GFP-SHIP164 using a double tagged cell RFP-Stx6 1-234 or full-length RFP-Stx6 Stx6 1 (Supplementary Figure 2A), however they didn't describe the goal of this panel in this context. Why were they used? It's clearer (the participation of Stx6) in another subheading (lines 312 -331) and its differences in presence and localization of SHIP164, regarding Stx6 overexpression.

12. Lines 381-383, describing Figure 6 B&C, the authors pointed to a decrease of large endosomes and it's also observed by representative images that EEA1 and GFP-2xHrsFYVE dots increase (independently of size) in SHIP164 KO cells. Is it possible to quantify? Are these changes/effects significant? If so, why these altered distribution and size occurs in those cells?

13. Lines 385-392, the authors described the localization of MPR, sortilin and TGN46 in SHIP164 KO cells and mentioned that the distribution was different in those cells compared to WT however another marker used, GM130 also seems different in KO cells (more concentrated closer to the nucleus). Did the analysis of spots/Golgi in Figure 6D-G include normalization or has there been an attempt to consider the effects of cis-Golgi morphological/distribution changes (GM130)?

14. In the subheading `Presence of Atg9 in clusters...` the authors claimed that SHIP164 could have a similar function of ATG2. Even though the overexpression of SHIP-164 and GFP-ATGA indicate proximity of the molecules, it's not showing that SHIP164 could act in lipid transfer to autophagosomes. The authors could provide more insights in SHIP164 function in autophagy if not only associate with Atg proteins or substitute functionally some molecules as ATG2. If the hypothesis of ATG2 and SHIP164 have similar role in lipid transfer, the downregulation or knockout of one of them should affect autophagosomes and ATG9A vesicles relative to autophagosomes.

15. Lines 447-450, the authors mentioned that no LC3 abundance changes were observed in SHIP164KO cells, which suggest that the protein has no a role on autophagy, but what analyses or assays were performed? Is any other autophagy associated protein evaluated in these cells? Is degradation of any known autophagy substrate affected?

16. LC3 is not a marker of autophagy only, and recent developments show that it is associated with various compartments (especially endocytic) that have nothing to do with authentic autophagy. The authors should avail themselves of these steady and now fully accepted developments and perform a more solid and up-to-data analysis of autophagy. This is critical since they deal with endosomal pathway and its interactions with the Golgi.

17. In the last subheading `UHRF1BP1, a SHIP164 paralogue, is also localized in endocytic pathway`, the authors addressed the paralogue as an endosomal marker (shown by previous work), whereas in this manuscript, seems more as a lysosomal one and it colocalizes with LAMP1. What are the differences between these paralogues that impact their function in the endocytic pathway? This appears to be left out hanging, and a good and orderly experimental approach is needed to make this comprehensible.

18. Line 459, describing exogenous ATG9A as also close to UHRF1BP1. Did the authors address this considering a latter role in endocytic vesicles. These issues, i.e. the relative roles of UHRF1BP1 and UHRF1BP1L (SHIP164) need to be disentangled, systematically addressed, and experimentally delineated, and hierarchically ordered.

Minor comments:

1. Lines 161 and 163, mention Figure 1 Ac-d, but there is no d in panel 1A. It's confusing. Also in Supplementary Figure 1. The capital and lower case letterers (e.g. "A" and "a" do not work well to keep tracking and definitely would not be distinguishable when spoken. Please find a different way to disambiguate the presentation.
2. The first paragraph of subheading `Presence of Atg9 in clusters...` (lines 424-435) seems more a discussion than an introduction to the goal and assay performed.
3. The pattern of the last phrase of the paragraph addressing the next subheading is confusing. A suggestion is to close a paragraph with a conclusion and move the phrase opening new questions from the paragraph ending to the first paragraph of each following subheading.
4. The authors should adopt a standard writing unit [for example, in line 1360, 48 hours; line 1391, 16hrs or in line 1157, *in vitro* and line 1161, *in vitro* (in italic)].
5. The references should be in Harvard style and listed alphabetically as described by JCB reference guidelines.
6. Line 958 (Figure legend 7A) the scale bar - units correct?
7. Line 975 (Figure legend 7D) the scale bar - units correct?
8. Line 1343 (Materials and Methods, Live Cell Imaging and Immunofluorescence) it`s written Halo-tag ligands ligands.
9. Lie 1382 (Materials and Methods, Mouse brain lysate...): C57/black mice, this is not the correct denomination of the strain.
10. In Supplemental Figure 6D-E there are blue dots around the nucleus, cytosolically dispersed. What does it mean?

Response to the reviewers. (*Authors responses are italicized*)

We would like to thank both reviewers for the many positive comments. We found the reviewers' suggestions on how to improve the manuscript very helpful. We also are grateful to the editors for allowing us to focus our revision on issues that are at the core of the scope of the paper while keeping experimentation on secondary objectives of the study for future further investigations. Topics on which we have not expanded, as agreed with the editors are:

- 1. A more detailed study of the SHIP164 paralogue UHRF1BP1: We thought it was important to mention that SHIP164 has a paralog. However, we consider a systematic characterization of this other protein a topic appropriate for a follow-up study. Thus, in the revised manuscript we simply focus on structural similarities with SHIP164 and its localization downstream to SHIP164 in the endocytic pathway (a Rab7 effector, rather than a Rab5 effector) and we have deleted the relation to ATG9, something that would have to be better developed.*
- 2. DYNLL1/2 and Rab45: A more systematic characterization and functional analysis of the interaction of the SHIP164 interaction with DYNLL1/2 and with Rab45 (two adaptors for the retrograde motor dynein). We believe that while the occurrence of these interactions validates a core message of the manuscript (a role of SHIP164 on retrograde vesicles), their systematic characterization is beyond the main scope of this manuscript.*

Detailed point-to point response to the reviewers' critiques.

Reviewer #1:

In this study, Hanna, Suen and colleagues examine the SHIP164 (UHRF1BP1L) protein which is a member of the Chorein lipid transfer protein family and is implicated in Parkinson's disease by a genome sequencing study. The authors perform convincing biochemical assays demonstrating that purified SHIP164 efficiently transfers lipids in vitro. Moreover, single particle cryo-electron microscopy experiments reveal that the SHIP164 structure resembles Chorein family members including VPS13 and ATG2, as recently predicted by AlphaFold. The authors also perform experiments to address the cellular function of SHIP164, suggesting roles in the trafficking of certain endosomal compartments from the cell periphery to perinuclear Golgi compartments and possibly autophagy. However, there are concerns that some of the functional tests rely on over-expression of the SHIP164 protein that results in indirect effects or artefacts. The authors take care to point out potential pitfalls in using the over-expressed protein, but then base some of their major conclusions on SHIP164 over-expression experiments. The purported association of SHIP164 with ATG9 vesicles is particularly egregious in this regard. Overall, this is an admirable study that incorporates biochemical, structural, and cell biological investigations, but additional tests are needed to substantiate some of the authors' conclusions regarding SHIP164 localization and function.

We would like to thank the reviewer for his/her thoughtful and pointed analysis of our manuscript. We were pleased to read that he/she found our study "admirable and containing convincing biochemical evidence of the lipid transfer capability of SHIP164.

Major concerns:

- 1. The authors identify DYNLL1/2 as candidate SHIP164 binding partners. However, a functional significance of this association is lacking in the study. The authors should examine endogenously tagged SHIP164 (eSHIP164^{mNG}) upon depletion of DYNLL1/2. One may also expect that depletion of DYNLL1/2 would result in phenotypes similar to those observed in SHIP164 knockout (KO) cells, including differences in endosome size as well as localization of the mannose 6-phosphate receptor (M6PR), TGN46, and sortilin. Is this the case? In reciprocal experiments, is the distribution of DYNLL1/2 altered in SHIP164 KO cells?*

Our loss-of-function studies suggest an important role for SHIP164 in the traffic between endosomes and the Golgi complex, a process partially dependent on the function of the dynein/dynactin complex of which DYNLL1/2 is known to assemble with. The biochemical interaction that we have discovered between

SHIP164 and DYNLL1/2, provides an additional piece of evidence linking SHIP164's function to retrograde traffic from the cell periphery to the center of the cell. This is why we believe that the report of such interaction enriches our paper.

However, the central idea put forward in our manuscript, that SHIP164 has a role in lipid dynamics in organelles of the endocytic pathway, is not dependent on a further understanding of this interaction. Furthermore, we are concerned about the interpretation of DYNLL1/2 KD or KO experiments as the putative consensus motif for DYNLL1/2 binding is present in >100 other proteins (PMID: 31266884) indicating that depletion of DYNLL1/2 would lead to pleiotropic effects in cells making difficult any interpretation of depletion experiments. Therefore, we chose not to pursue further the study of the SHIP164 - DYNLL1/2 interaction at this time.

2. The authors suggest that SHIP164 may be a Rab45 effector, but this is primarily based on experiments involving Rab45 over-expression. The authors should examine endogenously tagged SHIP164 (eSHIP164^{mNG}) upon depletion of Rab45. In addition, is M6PR distribution affected upon depletion of Rab45, similar to the SHIP164 KO cells?

As we discussed at the previous point for the binding of SHIP164 to DYNLL1/2, we find that evidence for a SHIP164-Rab45 interaction provides yet another piece of evidence linking SHIP164's function to retrograde traffic from the cell periphery to the center of the cell. Our findings demonstrate a major redistribution of both exogenous (Figure 5D) and endogenous (Supplemental Figure 3E) SHIP164 upon over-expression of Rab45, a protein with a well-established role in the retrograde traffic of endocytic vesicles. While we agree with the reviewer here that a Rab45 loss of function study would be an important complement of our study, such experiments are complicated by the existence of a Rab45 paralog, CRACR2a. Furthermore, involvement of both Rab45 and its paralog CRACR2a in retrograde traffic has already been established (PMID:30814157). Therefore, we believe that an in-depth analysis of Rab45 and its role in retrograde traffic would not advance the core findings of our manuscript.

3. The association of SHIP164 with ATG9 vesicles is tenuous. The authors found that over-expressed ATG9A-GFP is concentrated on SHIP164-positive foci upon SHIP164 over-expression (Figure 7B). Over-expression of SHIP164 along with Rab45 over-expression also increased the co-localization of these proteins with ATG9A (Figure 7C). However, since SHIP164 over-expression appears to induce "abnormal effects" upon endo-lysosomal compartments (Figure 3; lines 254-256), the over-expression data in Figure 7 does not 'prove' that ATG9A localizes to SHIP164-positive vesicles (lines 445-446). The authors should examine whether endogenously tagged SHIP164 (eSHIP164^{mNG}) co-localizes with ATG9A. The authors also observed that ATG9A is concentrated in the perinuclear Golgi complex and also present at compartments scattered throughout the cytoplasm (Figure 7A). Is the distribution of ATG9A altered in SHIP164 KO cells, similar to M6PR?

We thank the reviewer for raising these critical points that allowed us to make some important corrections in our manuscript. We have followed up on this comment and we have found little colocalization of endogenous SHIP164 with ATG9 (now shown in figure 7B). Thus, we have drastically downplayed the link of SHIP164 to ATG9 and we have removed reference to such link in the Abstract. Nevertheless, we have confirmed the striking colocalization of both endogenous (immunofluorescence) and exogenous (ATG9A-GFP) ATG9 (now shown in Figure 7C&D) with foci of overexpressed SHIP164. Importantly, we note that foci of exogenous SHIP164 show specificity for vesicles based on cargoes, as TGN46 is not affected by SHIP164 over-expression while MPR is dramatically enriched in such foci (Supplemental Figure 2D-I). Thus, the presence of ATG9 in these foci suggests some functional link between SHIP164 and ATG9 that will have to be further explored. We have accordingly modified the text as well as the title of the chapter of Results discussing this point. Old title: "Presence of ATG9A in clusters of SHIP164-positive vesicles". New Title: "Presence of ATG9A in vesicle clusters induced by SHIP164 overexpression"

Concerning the question of whether the distribution of ATG9A was altered in SHIP164 KO cells, we tested this possibility and we saw no effects (now shown in Supplemental Figure 5 B&C)

4. Likewise, the purported localization of the SHIP164 paralog UHRF1BP1 with ATG9A vesicles and LAMP1-positive lysosomes relies solely upon UHRF1BP1 over-expression experiments. Do the endogenous (or

endogenously tagged) UHRF1BP1 and ATG9A proteins co-localize? The authors should also examine ATG9 vesicle and lysosomal (LAMP1) distribution in cells depleted of UHRF1BP1 in order to substantiate their conclusions.

As both reviewers have considered some of our conclusions about the SHIP164 paralog UHRF1BP1 too preliminary, we have simplified this figure. We have eliminated some of the data on this protein, including (also in view of what we have discussed at the preceding point on ATG9 and SHIP164) the colocalization of overexpressed UHRF1BP1 with ATG9. However, we think it is important to include in this manuscript that SHIP164 has a paralog. Thus, in the revised manuscript we have included the overlaid AlphaFold predicted structure of SHIP164 and UHRF1BP1 to highlight their very similar structure (now shown in Figure 8A). Moreover, we show new panels better illustrating the colocalization of exogenous UHRF1BP1 with LAMP1 and Rab7 (Figure 8C&D). Unfortunately, we are not able to examine the endogenous localization of UHRF1BP1 at this time as the commercial antibodies against UHRF1BP1 have not been successful in immunocytochemical testing in cells and we have not been successful in endogenously tagging UHRF1BP1 in cells.

5. A possible role of ATG9 vesicles functioning in endo-lysosomal trafficking along with SHIP164/UHRF1BP1L and/or UHRF1BP1 seems like an interesting and potentially novel concept (although there may be some evidence for this notion in the published literature, PMID: 29156099). The authors may want to directly test whether ATG9 isoforms regulate endosome size as well as localization of M6PR, TGN46, and sortilin, just as they did for SHIP164. It is curious that ATG9A/B were not implicated in previous studies on endosomal sorting pathways (distinct from their roles in autophagy & beside PMID: 29156099). However, redundancies could explain why this has been missed or overlooked. To further test this model, the authors may want to examine whether loss of ATG9A or ATG9B is additive with loss of SHIP164/UHRF1BP1L or UHRF1BP1.

We have now downplayed the relation of SHIP164 and ATG9 (see above at point #3) and we will pursue a more in-depth analysis of this relation in follow-up studies.

Reviewer #2:

This is an “important and beautiful study addressing some of the cutting edge issues in membrane biogenesis and bulk lipid transfer independent of vesicular fusion”. The authors show that SHIP164, via its chorein motif, similar structurally to VPS13s and ATG2s, transfers bulk lipids. They show that it localizes to a cluster of small vesicles (presumed to be endocytic or of what origin?) with effects on endosomal-Golgi trafficking, mainly retrograde transport. Somewhat en passant/tangentially, an effect on autophagy was addressed since there is an unescapable similarity with ATG2s and apparent localization with ATG9A, etc. The hypothesis that SHIP164 is a lipid transport protein is supported by the results and is the main strength of this study, worth reporting. Should be of interest to many cell biologists.

However, it is necessary to strengthen the data behind the secondary objectives (past the lipid transfer) and improve the accessibility and interpretation of the results even concerning the primary objective. The absence/shortage of usable work with endogenous SHIP164 is a downside, but this plagues most of cell biology studies in general and is unavoidable. Similarly, in certain areas both the conceptual and technological approaches need to be improved/refined. In particular, the sections on autophagy need to be substantively and substantially revamped, and hierarchical issues regarding the roles of UHRF1BP1 and UHRF1BP1L need to be resolved. Nevertheless, this is an elegant and exciting study and this reviewer will be looking forward to seeing it in its revamped form should this happen.

We would like to thank the reviewer for defining our study “an important and beautiful study addressing some of the cutting edge issues in membrane biogenesis and bulk lipid transfer independent of vesicular fusion” and for the many suggestions on how to improve the paper.

Major comments:

1. A critical issue that needs to be addressed is the proposed role of SHIP164 in autophagy - knockout/mutational analysis is absolutely needed or else the whole connection (if at all) is circumstantial and substantially weakens the study.

Thank you for raising the important point regarding the relation between SHIP164 and autophagy. Based on several of the comments, we realized that our wording and presentation of the enrichment of ATG9A in SHIP164 clusters may have led to confusion about the function of SHIP164 that we propose. We did not intend to suggest that SHIP164 has a role in autophagy. In fact, we argue that our data suggests that the potential functional partnership of SHIP164 and ATG9A is evidence of a non-autophagy related role for ATG9A. We apologize for this confusion and have made significant efforts to make our point clearer in this version of the manuscript in the following ways:

1) We have amended the text in the chapter titled “Presence of ATG9A in clusters of SHIP164-positive vesicles” (now renamed “Presence of ATG9A in vesicle clusters induced by SHIP164 overexpression”) to make clear that we propose a non-autophagy related function of SHIP164.

2) We have added immunocytochemical staining demonstrating that endogenous ATG9A, but not the autophagy marker WIPI2 (mammalian Atg18), is enriched within SHIP164 foci demonstrating that these structures are not autophagosomes and cannot be implicated as an intermediate structure related to autophagy (now shown in Figure 7D). We also now show that exogenous SHIP164 accumulations do not co-localize with LC3 (RFP-LC3) under starvation or non-starvation conditions (now shown in Figure 7E).

3) We have added evidence that autophagic flux, as revealed by the conversion of LC3 (i.e. LC3-I to LC3-II), is not affected in SHIP164 loss-of-function cells (now shown in Figure 7F).

2. The aspect of lipid transfer and comparative model with ATG2s and VPS13s is unescapable and well-presented/used in this study. The point regarding ATG9A role outside of autophagy is exceptionally relevant and well taken. The authors may wish to elaborate a bit more. Because of the ATG2/ATG9A as comparators, it is critical to bring autophagy aspects of the study, currently severely curtailed to the current state of the field.

Thank you for emphasizing the importance of this point. SHIP164 and ATG2 share a rod-like structure that forms a hydrophobic groove (now shown in figure 7A) expected to transfer lipids between membrane bilayers. Newly reported roles for the non-autophagy related function of ATG9 (PMID: 34257406, 34799570, 35180289) led us to explore whether ATG9, having a significant endosomal distribution, could localize to endocytic membranes also positive for SHIP164. We do not suggest that the function of ATG2 is redundant with SHIP164 in autophagy, as discussed in response to comment #1.

3. Autophagosomes form without LC3 (PMID: 34741801), with multiple studies in support of that (22275429, 27864321, 31353311, 27885029, 32320309). Thus, equating LC3 with autophagy (used here as a key argument) is no longer acceptable (albeit perpetuated in lower quality studies). Furthermore, LC3 is associated with various compartments including endosomal ones. The authors should bring their analysis of autophagy to a much more contemporary level and with appropriate rigor and provide experimental details that are at present entirely lacking (there is only narrative but no data).

Once again, as mentioned in response to comment #1, we did not intend to suggest that SHIP164 functions in autophagy. We have added new data demonstrating that while ATG9A is enriched in SHIP164 foci, such structures are distinct from autophagic organelles as WIPI2, an essential autophagy protein, is not present in these clusters (now shown in Figure 7D).

We appreciate the reviewer’s hesitancy in placing too much importance on the localization of LC3 as a readout for autophagy. Therefore, we compared the conversion of LC3 (i.e. LC3-I to LC3-II) in wild-type and SHIP164 knockout cells and determined that the loss of SHIP164 function does not impair autophagic flux in these cells compared with wild-type. While this assay again relies on LC3 as a readout, it has been successfully used to examine the impact of ATG2 loss-of-function in mammalian cells (PMID: 30952800) (now shown in Figure 7F).

4. Another key issue relates to UHRF1BP1 and UHRF1BP1L (SHIP164). The roles and relationships between these two paralogues and their inter-related roles in processes need to be disentangled, systematically addressed experimentally and hierarchically ordered within the pathways studied.

We thought that it was important to mention that SHIP164 has a paralog, but we also thought that a systematic analysis of this paralog was beyond the scope of this already data-heavy study. Thus, we concluded the Results section with short paragraph mentioning its existence and some preliminary data indicating similarities to, and differences from, SHIP164. Based on the comments of both reviewers we have removed the colocalization of overexpressed UHRF1BP1 with ATG9, added a new panel better showing the localization of exogenous UHRF1BP1 next to LAMP1 and Rab7-positive vesicles (now shown in Figure 8C&D) and added a structural comparison based on AlphaFold (now shown in Figure 8A). We feel that an extensive evaluation of UHRF1BP1 is better suited for follow up studies.

5. The last sentence of the abstract seems circular, rather than providing an expected clear conclusion. It reads as if the bulk lipid transfer plays a role in endosomal trafficking, whereas it is probably meant to say that endosomal trafficking plays a role in bulk lipid transfer. Even so, in this latter meaning, how does that help with the main premise, since the whole story is built on bulk lipid transfer independently of vesicular traffic, as stated? Perhaps the authors may want to recast this whole semantic exercise in the context of differentiating "vesicular fusion" rather than "vesicle traffic". A potentially good statement is found in the discussion, and would work well in the abstract (paraphrased/modified) : "controlling lipid transfer between membrane at the interface with vesicular trafficking" or a variant.

We are grateful to the reviewer for the suggestion to improve the Abstract, but we wonder whether the meaning of the last sentence of the Abstract may have been misinterpreted. We think that the old sentence "Our findings raise the possibility that bulk transfer of lipids to endocytic membranes may play a role in their traffic" conveys the message that we wanted to communicate, i.e. that bulk lipid transport by SHIP164 may serve to control lipids and protein-lipid ratio in the membranes of endocytic vesicles and that this, in turn, may impact traffic within the endocytic pathway. We are not sure that the sentence suggested by the reviewer would make the point clearer and we could not come up with a better sentence.

6. The shotgun lipidomics of lipids associated with the purified SHIP164 is very important and elegant. However, the summary representation in Fig. 1B is not satisfactory. Where are the primary data and if so can they be tabulated per exact lipid species (rather than the bland PC, PS, PI bar graphs? Were there any other lipids except phospholipids? How about cholesterol and sphingolipids? If yes, ratios to phospholipids (dis-enrichment) , if not, how is that interpreted?

The tabulated lipidomics data is now available in the source data file for Figure 1 uploaded with the revision material. The purpose of the lipidomics analysis of purified SHIP164 was to confirm its ability to harbor glycerolipids in an aqueous environment. Consistent with other VPS13/ATG2 proteins, we did not uncover a striking enrichment of a particular phospholipid.

7. It is confusing to first follow a lot of analyses with overexpressed SHIP164 fusions and then to learn that the authors conclude that much of that is an "over-expression defect" (line 309). Are all observations near the Golgi then an artifact? It would be good for authors to recast this work separating the "exploratory" stages, i.e. what belongs in notebooks and possibly can be all placed in a single supplementary figure illustrating some of those relationships that are per authors' own statement "artifactual", from "conclusive data". In other words, while some of the "historically" presented explorations are pretty it is disappointing to reach a conclusion that all we've been following is an artifact to a certain point. Please separate wheat from the chaff.

Thank you for this comment. We had thought (and even more so after your comment) to start with the localization of the endogenously tagged protein. However, the studies with the overexpressed protein have provided important data that have then guided us in the analysis of the endogenously expressed protein. Thus, we have decided to keep the original flow of the paper. Building on your comment, we have deleted some non-essential data from this section and rephrased some sentences to emphasize the exploratory nature of this first part of the study.

We removed the following data:

1) Removed Supplemental Figure 2A: *This data confirms that GFP-SHIP164 (i.e. SHIP164 tagged at the N-terminus) is only recruited to membrane when full-length Stx6 is co-expressed. We included these*

data to demonstrate reproducibility of previous findings, but it is redundant with already published work (PMID: 20163565) and therefore have removed this data.

2) Removed Supplemental Figure 2B: This data was to demonstrate the dynamic nature of exogenous SHIP164 (SHIP164^{mScarlet}) juxtaposed on macropinosomes marked with GFP-WDFY2. This data is redundant with fluorescence images in Figure 3 and has been removed.

3) Removed Supplemental Figure 2D: This data demonstrates that the MPR is enriched on exogenous SHIP164^{mScarlet} foci in HeLa cells. We included these data to demonstrate the clustering effect of SHIP164 over-expression in multiple cell lines, but it is redundant with Figure 2E.

4) Removed Supplemental Figure 3: A truncated form of SHIP164 (SHIP164₁₋₈₇₃-mRFP) can form large foci when over-expressed in cells. We have removed this data due to its speculative nature about the presence of exogenous SHIP164 and the formation of larger vesicle clusters.

We moved the following data to the supplement:

1) Moved Figure 2B to Supplemental Figure 2A. This data demonstrates that a small tag engineered to the N-terminus of SHIP164 (3xFLAG-SHIP164^{mScarlet}) excludes the formation of exogenous SHIP164 accumulations.

2) Moved Figure 2C to Supplemental Figure 2B. This data was the first evidence that exogenous SHIP164^{mScarlet} localized to accumulations juxtaposed on endosomal membrane labeled with the PI3P marker GFP-Hrs^{FYVE}.

3) Moved Figure 2H&I to Supplemental Figure 2D&E. This data demonstrates that untagged exogenous SHIP164 induces the formation of accumulations enriched for the MPR ruling out artifacts of the fluorescently tagged SHIP164 in other experiments (i.e. SHIP164^{mScarlet}).

8. If ATG9A is found on "foci" resulting from overexpression, how is one to interpret this given the point re "overexpression defect" above?

We have addressed this comment at point #3 of reviewer #1.

9. Where are the complete MS data for SHIP164 interacting partners? Only cherry-picked dynein "data" are shown in Fig 5B.

The original mass-spec data is now available in the source data file for Figure 5 uploaded with the revision material.

10. It doesn't seem clear if the full-length construct of SHIP164 is prone to aggregate in all conditions and assays of the manuscript, especially in Figure 1. A related suggestion is to state in figure legends what and when the authors used (e.g. the 3xFLAG-SHIP164 Δ 901-1099).

Aggregation was found to be an issue only during cryo-EM analysis after purification, which was remedied with the indicated truncation SHIP164 Δ 901-1099 and used for all lipid transfer and structural analysis studies. We have added clearer labels to Figure 1 indicating what construct was used and when. Thank you for raising this point.

11. In the first paragraph of subheading `Localization of exogenous SHIP164...` (lines 172-176), the authors describe the localization of GFP-SHIP164 using a double tagged cell RFP-Stx6 1-234 or full-length RFP-Stx6 Stx6 1 (Supplementary Figure 2A), however they didn't describe the goal of this panel in this context. Why were they used? It's clearer (the participation of Stx6) in another subheading (lines 312 -331) and its differences in presence and localization of SHIP164, regarding Stx6 overexpression.

We confirmed the localization of SHIP164 tagged at the N-terminus (GFP-SHIP164) previously reported (PMID:20163565) to be cytosolic by co-expressing GFP-SHIP164 with either full-length Stx6 (mRFP-Stx6)

or a soluble Stx6 lacking the C-terminal transmembrane domain (mRFP-Stx6₁₋₂₃₄). We found, as expected, that the full-length Stx6 recruited GFP-SHIP164 to membranes of the Golgi complex and endosomal structures while GFP-SHIP164 was cytosolic in the presence of the soluble Stx6. This demonstrates that GFP-SHIP164 is cytosolic and only recruited to membranes in the presence of exogenous full-length Stx6. We hypothesized that the N-terminus of SHIP164 is sensitive to engineered tags and decided to identify internal sites to engineer fluorescent tags used throughout this manuscript. Internal tagging of SHIP164 within the disordered region was essential for our endogenous localization experiments. However, while these experiments were critical in our localization studies, we acknowledge that they are redundant with published work and have removed Supplemental Figure 2A (also in discussed in response to comment #7 of efforts to focus early experiments using SHIP164 over-expression).

12. Lines 381-383, describing Figure 6 B&C, the authors pointed to a decrease of large endosomes and it's also observed by representative images that EEA1 and GFP-2xHrsFYVE dots increase (independently of size) in SHIP164 KO cells. Is it possible to quantify? Are these changes/effects significant? If so, why these altered distribution and size occurs in those cells?

In lines 381-383 of the original manuscript, we describe only our finding that the largest endosomes that are identified using immunocytochemical staining against EEA1 are absent in cells lacking SHIP164. We do not claim to have found a change in the number of EEA1 positive foci independent of size in our quantification panels in Figure 6B and Supplemental Figure 6G (now Supplemental Figure 4G).

13. Lines 385-392, the authors described the localization of MPR, sortilin and TGN46 in SHIP164 KO cells and mentioned that the distribution was different in those cells compared to WT however another marker used, GM130 also seems different in KO cells (more concentrated closer to the nucleus). Did the analysis of spots/Golgi in Figure 6D-G include normalization or has there been an attempt to consider the effects of cis-Golgi morphological/distribution changes (GM130)?

We did not observe an obvious difference in the morphology/distribution of GM130 in our SHIP164 knockout cells compared with wild-type cells. Our quantification of the ratio of signal from cytoplasmic spots to the signal in the Golgi complex using immunocytochemical staining uses GM130 staining, but not intensity, to differentiate between non-Golgi complex cytoplasm and the Golgi complex.

14. In the subheading 'Presence of Atg9 in clusters...' the authors claimed that SHIP164 could have a similar function of ATG2. Even though the overexpression of SHIP-164 and GFP-ATGA indicate proximity of the molecules, it's not showing that SHIP164 could act in lipid transfer to autophagosomes. The authors could provide more insights in SHIP164 function in autophagy if not only associate with Atg proteins or substitute functionally some molecules as ATG2. If the hypothesis of ATG2 and SHIP164 have similar role in lipid transfer, the downregulation or knockout of one of them should affect autophagosomes and ATG9A vesicles relative to autophagosomes.

We did not intend to state that SHIP164 and ATG2 could have a similar function in autophagosome formation. The similarity to which we refer is the structural similarity and the shared property to transfer lipids revealed by an in vitro assay, but we hypothesize that SHIP164 has different functions in cell physiology: ATG2 in autophagy and SHIP164 in the endocytic pathway. We have now tried to make this clearer in the paragraph under the subheading "Presence of ATG9A in vesicle clusters induced by SHIP164 overexpression" (formerly titled Presence of ATG9A in clusters of SHIP164-positive vesicles). The new data included in the revised manuscript, presented in response to your comment #1 and #3, clearly speaks against a role of SHIP164 in autophagy.

15. Lines 447-450, the authors mentioned that no LC3 abundance changes were observed in SHIP164KO cells, which suggest that the protein has no a role on autophagy, but what analyses or assays were performed? Is any other autophagy associated protein evaluated in these cells? Is degradation of any known autophagy substrate affected?

In the updated version of our manuscript, we have included new data demonstrating that the loss of SHIP164 does not impact autophagy. This was also presented in response to your comments #1, #3, and #14. Briefly, we show that the conversion of LC3, a readout successfully used to measure the effect of ATG2

protein loss-of-function on autophagy (PMID: 30952800), was not altered in cells lacking SHIP164 (now shown in Figure 7F).

16. LC3 is not a marker of autophagy only, and recent developments show that it is associated with various compartments (especially endocytic) that have nothing to do with authentic autophagy. The authors should avail themselves of these steady and now fully accepted developments and perform a more solid and up-to-date analysis of autophagy. This is critical since they deal with endosomal pathway and its interactions with the Golgi.

We agree with the reviewer that autophagy is a complex process that may have many different readouts. As we have done in response to your comment #1, we would like to emphasize that we do not suggest a function of SHIP164 related to autophagy. Therefore, we believe that a systematic analysis of the effects of SHIP164 loss-of-function on autophagy to not be in the purview of our current study.

17. In the last subheading `UHRF1BP1, a SHIP164 paralogue, is also localized in endocytic pathway`, the authors addressed the paralogue as an endosomal marker (shown by previous work), whereas in this manuscript, seems more as a lysosomal one and it colocalizes with LAMP1. What are the differences between these paralogues that impact their function in the endocytic pathway? This appears to be left out hanging, and a good and orderly experimental approach is needed to make this comprehensible.

We do not suggest that UHRF1BP1 is an endosomal marker, but rather that, like SHIP164, it localizes in foci juxtaposed to endolysosomes. However, while SHIP164 foci are juxtaposed to structures positive for early endosomal markers, UHRF1BP1 is juxtaposed to structures positive for LAMP1. This fits with the previous study (PMID: 31294692) reporting SHIP164 in the interactome of Rab5 and UHRF1BP1 in the interactome of Rab7. We have included in the updated manuscript (also presented in response to your comment #4) new panels better showing the colocalization of exogenous UHRF1BP1 with LAMP1 and Rab7, which can be found in what is now Figure 8C&D. We have now also stated and shown that UHRF1BP1 foci undergo fission (Figure 8E) suggesting that, like SHIP164 foci, they are represented by clusters of vesicles.

18. Line 459, describing exogenous ATG9A as also close to UHRF1BP1. Did the authors address this considering a latter role in endocytic vesicles. These issues, i.e. the relative roles of UHRF1BP1 and UHRF1BP1L (SHIP164) need to be disentangled, systematically addressed, and experimentally delineated, and hierarchically ordered.

We deleted the data concerning the proximity of ATG9 to UHRF1BP1.

Minor comments:

1. Lines 161 and 163, mention Figure 1 Ac-d, but there is no d in panel 1A. It's confusing. Also in Supplementary Figure 1. The capital and lower case letterers (e.g. "A" and "a" do not work well to keep tracking and definitely would not be distinguishable when spoken. Please find a different way to disambiguate the presentation.

Thank you for pointing out this error. We have corrected the text accordingly.

2. The first paragraph of subheading `Presence of Atg9 in clusters...` (lines 424-435) seems more a discussion than an introduction to the goal and assay performed.

We have modified this section extensively as discussed above.

3. The pattern of the last phrase of the paragraph addressing the next subheading is confusing. A suggestion is to close a paragraph with a conclusion and move the phrase opening new questions from the paragraph ending to the first paragraph of each following subheading.

We have considered this change but decided not to follow-up on this point which the reviewer considers a minor issue.

4. The authors should adopt a standard writing unit [for example, in line 1360, 48 hours; line 1391, 16hrs or in line 1157, in vitro and line 1161, in vitro (in italic)].

Thank you for pointing out this inconsistency in our units. We have changed the text accordingly.

5. The references should be in Harvard style and listed alphabetically as described by JCB reference guidelines.

The format of the references has been changed to JCB standards.

6. Line 958 (Figure legend 7A) the scale bar - units correct?

Thank you for pointing out this error. We have changed the text accordingly.

7. Line 975 (Figure legend 7D) the scale bar - units correct?

Thank you for pointing out this error. We have changed the text accordingly.

8. Line 1343 (Materials and Methods, Live Cell Imaging and Immunofluorescence) it`s written Halo-tag ligands ligands.

Thank you for pointing out this error. We have changed the text accordingly.

9. Line 1382 (Materials and Methods, Mouse brain lysate...): C57/black mice, this is not the correct denomination of the strain.

Thank you for pointing out this error. We have changed the text accordingly.

10. In Supplemental Figure 6D-E there are blue dots around the nucleus, cytosolically dispersed. What does it mean?

The ectopic blue signal is due to autofluorescence from 405 laser used to image the dim DAPI signal.

March 29, 2022

RE: JCB Manuscript #202111018R

Dr. Pietro V De Camilli
Yale University
Cell Biology and Neuroscience
295 Congress Avenue, BCMM 236
Lab BCMM 235-237
New Haven, CT 06519-1418

Dear Dr. De Camilli:

Thank you for submitting your revised manuscript entitled "SHIP164 is a Chorein Motif Lipid Transfer Protein that Controls Endosome-Golgi Membrane Traffic". We would be happy to publish your paper in JCB pending final revisions necessary to meet our formatting guidelines (see details below).

A. MANUSCRIPT ORGANIZATION AND FORMATTING:

- 1) Text limits: Character count for Articles is < 40,000, not including spaces. Count includes abstract, introduction, results, discussion, and acknowledgments. Count does not include title page, figure legends, materials and methods, references, tables, or supplemental legends.
- 2) Figures limits: Articles may have up to 10 main text figures.
- 3) Figure formatting: Scale bars must be present on all microscopy images, including inset magnifications. Molecular weight or nucleic acid size markers must be included on all gel electrophoresis.
- 4) Statistical analysis: Error bars on graphic representations of numerical data must be clearly described in the figure legend. The number of independent data points (n) represented in a graph must be indicated in the legend. Statistical methods should be explained in full in the materials and methods. For figures presenting pooled data the statistical measure should be defined in the figure legends. Please also be sure to indicate the statistical tests used in each of your experiments (either in the figure legend itself or in a separate methods section) as well as the parameters of the test (for example, if you ran a t-test, please indicate if it was one- or two-sided, etc.). Also, if you used parametric tests, please indicate if the data distribution was tested for normality (and if so, how). If not, you must state something to the effect that "Data distribution was assumed to be normal but this was not formally tested."
- 5) Abstract and title: The abstract should be no longer than 160 words and should communicate the significance of the paper for a general audience. The title should be less than 100 characters including spaces. Make the title concise but accessible to a general readership.
- 6) Materials and methods: Should be comprehensive and not simply reference a previous publication for details on how an experiment was performed. Please provide full descriptions in the text for readers who may not have access to referenced manuscripts.
- 7) Please be sure to provide the sequences for all of your primers/oligos and RNAi constructs in the materials and methods. You must also indicate in the methods the source, species, and catalog numbers (where appropriate) for all of your antibodies. Please also indicate the acquisition and quantification methods for immunoblotting/western blots.
- 8) Microscope image acquisition: The following information must be provided about the acquisition and processing of images:
 - a. Make and model of microscope
 - b. Type, magnification, and numerical aperture of the objective lenses
 - c. Temperature
 - d. Imaging medium
 - e. Fluorochromes
 - f. Camera make and model

g. Acquisition software

h. Any software used for image processing subsequent to data acquisition. Please include details and types of operations involved (e.g., type of deconvolution, 3D reconstitutions, surface or volume rendering, gamma adjustments, etc.).

10) Supplemental materials: There are strict limits on the allowable amount of supplemental data. Articles may have up to 5 supplemental figures. Please also note that tables, like figures, should be provided as individual, editable files. A summary of all supplemental material should appear at the end of the Materials and methods section.

13) ORCID IDs: ORCID IDs are unique identifiers allowing researchers to create a record of their various scholarly contributions in a single place. At resubmission of your final files, please consider providing an ORCID ID for as many contributing authors as possible.

Please note that JCB now requires authors to submit Source Data used to generate figures containing gels and Western blots with all revised manuscripts. This Source Data consists of fully uncropped and unprocessed images for each gel/blot displayed in the main and supplemental figures. Since your paper includes cropped gel and/or blot images, please be sure to provide one Source Data file for each figure that contains gels and/or blots along with your revised manuscript files. File names for Source Data figures should be alphanumeric without any spaces or special characters (i.e., SourceDataF#, where F# refers to the associated main figure number or SourceDataFS# for those associated with Supplementary figures). The lanes of the gels/blots should be labeled as they are in the associated figure, the place where cropping was applied should be marked (with a box), and molecular weight/size standards should be labeled wherever possible.

B. FINAL FILES:

Thank you for this interesting contribution, we look forward to publishing your paper in Journal of Cell Biology.

Sincerely,

Mark von Zastrow, MD, PhD
Monitoring Editor

Andrea L. Marat, PhD
Senior Scientific Editor

Journal of Cell Biology

Reviewer #1 (Comments to the Authors (Required)):

This study is significantly improved and my comments and concerns have been addressed by new experiments, revisions to the manuscript, and reasonable arguments in the authors' response letter. The study brings new insight into the role of lipid transfer proteins in membrane trafficking.

Reviewer #2 (Comments to the Authors (Required)):

The authors have chosen to address certain comments while choosing not to address others. In principle, the study was excellent to start with. Some of the amendments here clarify issues and shore up the central story.